# Learning to Select Exogenous Events
# for Marked Temporal Point Process

**Ping Zhang**[*]     **Rishabh Iyer**[*]     **Ashish Tendulkar**[†]     **Gaurav Aggarwal**[†]     **Abir De**[§]

[*]UT Dallas, [†]Google Inc., [§] IIT Bombay.
{Ping.Zhang,rishabh.iyer}@utdallas.edu,
{ashishvt,gauravaggarwal}@google.com, abir@cse.iitb.ac.in

## Abstract

Marked temporal point processes (MTPPs) have emerged as a powerful modeling tool for a wide variety of applications which are characterized using discrete events localized in continuous time. In this context, the events are of two types— *endogenous events* which occur due to the influence of the previous events and *exogenous events* which occur due to the effect of the externalities. However, in practice, the events do not come with endogenous or exogenous labels. To this end, our goal in this paper is to identify the set of exogenous events from a set of unlabeled events. To do so, we first formulate the parameter estimation problem in conjunction with exogenous event set selection problem and show that this problem is NP hard. Next, we prove that the underlying objective is a monotone and $\alpha$-submodular set function, with respect to the candidate set of exogenous events. Such a characterization subsequently allows us to use a stochastic greedy algorithm which was originally proposed in [64] for submodular maximization. However, we show that it also admits an approximation guarantee for maximizing $\alpha$-submodular set function, even when the learning algorithm provides an imperfect estimates of the trained parameters. Finally, our experiments with synthetic and real data show that our method performs better than the existing approaches built upon superposition of endogenous and exogenous MTPPs.

## 1 Introduction

In recent years, marked temporal point processes (MTPPs) have shown tremendous potential in modeling the arrival process of the asynchronous events in continuous time [65, 46, 66, 16, 22, 17, 52, 82, 33, 13, 60, 83, 74, 49, 77, 27, 53, 47]. They are extensively used in a wide variety of applications, *e.g.*, information diffusion in social networks [66, 77, 16, 43, 12, 85, 76], improving human learning [76, 74], location modeling [83], epidemic forecasting [49, 66], etc. At the outset, an MTPP event consists of two elements— its arrival time and the mark, where the latter encodes the category of the event, *e.g.*, the topic of a blog post, the sentiment associated with a tweet, disease of an individual, *etc*.

In the context of learning an MTPP, we observe two types of events— *endogenous events* which occur due to influence of the previously observed events and, *exogenous events* which are driven by the external sources or unobserved events, rather than being influenced by the previously observed events. For example, a post in social media may be influenced by some external news source, rather than the previous messages [26, 60, 13]; in epidemic process, an infection may be triggered by community transmission where the source is other than any previously known infections [50], etc. In such applications, the observed events appear unlabeled, *i.e.*, they are not tagged with their sources of influence. Consequently, the existing MTPP models [21, 12, 84, 78, 66] resort to the superposition of exogenous and endogenous MTPPs. However, such approaches view an event being exogenous or endogenous with equal likelihood and therefore, they do not focus on identifying which events are more likely to be exogenous than others. Consequently, they often remain oblivious to the presence of the exogenous events, which often preclude these models from realizing their full potential. While

35th Conference on Neural Information Processing Systems (NeurIPS 2021).

De et al. [13] tackle this problem in the particular context of opinion dynamics in social networks, their formulation focus only on the mark dynamics and, skirt the effect of exogenous events on the arrival times. As a result, their framework can not be extended to a generic MTPP model, where the arrival time is often a key component.

## 1.1 Our contributions

In order to respond to the above limitations, we make the following contributions.

**Learning in conjunction with selection of exogenous events.** We design TPP-SELECT, a novel event selection algorithm which, for any given MTPP model, simultaneously filters the exogenous events from a stream of unlabeled events as well as train the model parameters over the remaining (endogenous) events (Section 3). More specifically, we seek to maximize the likelihood of all the events, jointly with respect to the set of exogenous events and the parameters of the associated MTPP model. Such a formulation induces a parameter estimation problem in conjunction with the combinatorial optimization problem for selecting the exogenous events.

**Characterization of the objective.** Subsequently, we show that our problem is NP-hard due to the involvement of the candidate exogenous events as an optimization variable. We tackle this problem using several technical innovations. First, we reformulate it as an instance of a set function maximization problem, where the objective corresponds to the value of the maximum likelihood, as a function of the candidate set of exogenous events. Then, based on this representation, we show that this objective satisfies monotonicity and $\alpha$-submodularity— a notion of approximate submodularity [25, 44, 29] for linear MTPP and a wide variety of nonlinear MTPPs (Section 4). These results reveal an unexplored connection between approximate submodularity and temporal point process, that is of independent interest.

**Approximation guarantee for greedy method in the face of imperfect training.** Next, we show that these properties allow us to use an existing stochastic greedy algorithm [64]. While this algorithm is known to admit an approximation guarantee for submodular maximization, we show that it also enjoys an approximation guarantee for maximizing $\alpha$-submodular functions. Moreover, we show that this algorithm also admits an approximation guarantee even when the training algorithm provides an imperfect estimation of the MTPP parameters.

Finally, we perform a comprehensive evaluation of our proposal on both synthetic and real data. Our experiments with synthetically generated sequences show that TPP-SELECT can accurately select the exogenous events from a stream of unlabeled events, even though it is completely unsupervised. Our experiments on real datasets show that, despite training on a subset (endogenous) of events, TPP-SELECT outperforms several baselines as well as the corresponding model which is trained on the entire dataset in the presence of exogenous events. Our code is available in https://github.com/noilreed/TPP-Select.

## 2 Related work

Apart from the works on MTPPs, our work is related to robust learning, subset selection, modeling external effects in different applications, active learning, etc.

**Robust learning.** Robust learning focuses on training machine learning models in the presence of outliers and noisy labels [9, 67, 3, 23, 88, 4, 58]. However, exogenous events are not outlier data, they are events which follow a different model, other than the central model— which, in our case, captures the endogenous event dynamics. These robust learning methods predominantly consider i.i.d. data and focus on designing models and training algorithms which would generalize well across the data including the outliers. In contrast, the events in our setup have strong temporal dependencies. Moreover, our objective is to select the exogenous events, so that the model can be trained on the remaining (endogenous) events to obtain more accurate parameter estimates

**Data subset selection.** In recent years, there is a flurry of work on data selection [81, 80, 79, 48, 2, 40, 51, 57, 7, 6, 36], which predominantly select a smaller sized subset of data to facilitate efficient training, by optimizing submodular functions and their variants. However, TPP-SELECT aims to filter the exogenous events from a sequence of events to help the underlying MTPP model achieve better predictive performance.

**Modeling external effects.** Our work is also connected to the methods that model the external effect in several applications, especially, in web or online social media [19, 60] that aim to model the effect of exogenous influence on the information diffusion in social media. However, they do not aim to select the exogenous messages from a stream of unlabeled messages. Iwata et al. [34] aim to learn

the effect of one user on the post of other users. Such a model can be used to quantify the external effect. Linderman and Adams [45] aim to learn a probabilistic model which captures latent influence present in data using mutually exciting processes combined with a random graph model. Furthermore it infers the latent network structure from the noisy observations. Tanaka et al. [75] model dynamics of event sequence in the particular application of purchase data modeling. Specifically, it combines three factors, namely user preferences, influence from other users and media advertisements. These proposals learn their model using EM algorithm combined with Gibbs sampling method. However, they do not provide an algorithm to explicitly demarcate between endogenous and exogenous events. Very recently, Liu and Hauskrecht [47] consider a semi-supervised setting of event outlier detection for MTPP. However, they assume that they have access to data without outliers, so that the underlying MTPP model can be trained on this clean data. In contrast, our work applies to an unsupervised setting, where the training data also contains the exogenous events.

**Optimization of submodular functions and their variants.** Submodular functions have the ability to characterize a wide variety of set functions. Moreover, thanks to the existence of a simple greedy algorithm to maximize such functions, their use is widespread across several subset selection tasks [81, 80, 79, 48, 2]. Several popular set functions, *e.g.*, facility location, set cover, concave over modular, etc. are submodular [36, 2, 35]. In this context, there is a recent line of works which focus on maximizing different variants of submodular functions which include $\alpha$-submodular functions and $\gamma$-weakly submodular functions [61, 54, 5, 11, 37]. In particular, our work is more focused around $\alpha$-submodular functions.

**Subset selection in conjunction with model learning.** Existing works [7, 51, 36, 39, 14, 15, 57] have also considered subset selection in conjunction with model parameter estimation. However, in the context of applications as well as formulations, these works differ from us. Killamsetty et al. [39] consider simultaneous parameter estimation on training set and subset selection using validation error in order to facilitate efficient learning without introducing significant error. On the other hand, the works in [36, 51, 57, 7] focus on selecting coreset— a small representative subset of the entire data— together with parameter estimation. However, they consider i.i.d data and select subset for efficient training. In contrast, we partition sequence of events into exogenous and endogenous events for improved training of point process models. Finally, the works in [14, 15] consider simultaneous parameter estimation and subset selection. Similar to our work, they aim to find model parameters and a desired data subset simultaneously by minimizing the underlying loss functions with respect these variables. However, they focus on human assisted machine learning, whereas our work considers exogenous event selection in the context of marked temporal point processes. Thus the problem formulation, results and proofs are significantly different from these works.

**Active learning.** Active learning methods [81, 28, 69, 42, 24] aim to identify instances in the training data, in order to reduce the labelling cost. In contrast, our work aims to select samples which are not influenced by the previous events in a sequence of dependent observations. To that end, it trains the endogenous MTPP model on the rest of the events. More specifically, active learning methods aim to find a subset of instances so that the model trained over it gives accurate predictions for *all the samples* in the test set. In contrast, our work selects both exogenous and endogenous events and then trains two separate models $(\lambda_{\boldsymbol{\theta}}, m_{\boldsymbol{\theta}})$ and $(\mu, q)$ so that they perform well specifically on endogenous and exogenous events respectively.

## 3  Problem formulation

In this section, we formulate our problem of learning an MTPP in presence of exogenous events, starting with an overview of MTPPs.

### 3.1  Overview of marked temporal point processes

A marked temporal point process is a stochastic process which is characterized using a sequence of events $\mathcal{H} = \{e = (y, t)\}$, where $t \in \mathbb{R}^+$ is the time of occurrence and $y \in \mathcal{C} = \{c\}$ is mark of the event that has happened at time $t$ [17, 12, 65, 10]. The dynamics of the arrival times can equivalently be represented as a counting process $N(t)$ that counts the number of events occurred until time $t$. In this context we also define $\mathcal{H}_t$ as the history of events happened until and excluding time $t$, *i.e.* $\mathcal{H}_t = \{e_i = (y_i, t_i) | e_i \in \mathcal{H}, t_{i-1} < t_i < t\}$. We specify the dynamics of the arrival times using a conditional intensity function $\lambda(t)$ which in turn encapsulates the probability that an event occurs in the infinitesimal time interval $[t, t + dt)$, described as follows [12, 74, 10]:

$$\lambda(t) = \Pr\left(\text{An event will occur between } t \text{ and } t + dt\right) = \mathbb{E}[dN(t) = 1], \qquad (1)$$

where $dN(t) \in \{0, 1\}$ is the number of events arriving in the infinitesimal time-interval $[t, t + dt)$. On the other hand, we capture the behavior of marks using a probability distribution, *i.e.*

$$m(c) = \Pr(\text{Mark of an event is } c) = \Pr(y = c), \tag{2}$$

which is often modeled as a multinomial distribution over the set of marks $\mathcal{C}$.

## 3.2 Problem setup

**Endogenous and exogenous events.** In the context of MTPP, we observe two types of events, viz., (i) endogenous events that occur due to the influence of the previously observed events; and, (ii) exogenous events that are *not* influenced by the previously observed events but occur due to the influence of the externalities [21, 13]. The dynamics of the endogenous events are modeled using a parameterized conditional intensity function $\lambda(t) = \lambda_{\boldsymbol{\theta}}(t \,|\, \mathcal{H}_t)$ and conditional mark distribution $m(y) = m_{\boldsymbol{\theta}}(y \,|\, \mathcal{H}_t)$. The arrival time and the mark depend on the previously observed events $\mathcal{H}_t$,

$$\lambda_{\boldsymbol{\theta}}(t \,|\, \mathcal{H}_t) = \mathbb{E}(dN(t) = 1 \,|\, \mathcal{H}_t), \quad m_{\boldsymbol{\theta}}(y \,|\, \mathcal{H}_t) = \Pr(y \,|\, \mathcal{H}_t). \tag{3}$$

Therefore, both $\lambda_{\boldsymbol{\theta}}$ and $m_{\boldsymbol{\theta}}$ are stochastic quantities. On the other hand, exogenous events are captured mostly using an independent MTPP in which, the intensity function or the mark distribution are modeled using a deterministic and mostly time invariant distributions. In our work, we model the intensity function and the mark distribution for exogenous events as Gamma and Beta noise,

$$\lambda(t) = \mu \sim \text{Gamma}(\alpha_\lambda, \beta_\lambda), \qquad m(y) = q \sim \text{Beta}(\alpha_{m,y}, \beta_{m,y}), . \tag{4}$$

These distributions allows $\mu \in (0, \infty)$ and $q \in (0, 1)$. $\alpha_\bullet$ and $\beta_\bullet$ are hyperparameters. We further assume that both the arrival times and the marks of the exogenous events are *i.i.d.* random variables.

**Use of superposition of MTPP models and their limitations.** Suppose, if at all possible, the training observations $\mathcal{H}_T$ gathered during the time-window $[0, T)$ were already tagged with endogenous or exogenous labels. In such a case, one could first build the set of exogenous events $\mathcal{S}$ and the endogenous events $\mathcal{H}_T \backslash \mathcal{S}$ and then, use maximum likelihood estimation to learn $\{\alpha_\bullet, \beta_\bullet\}$ and $\boldsymbol{\theta}$ separately using the set of exogenous events $\mathcal{S}$ and the endogenous events $\mathcal{H}_T \backslash \mathcal{S}$, respectively.

In practice, the difference between the exogenous and the endogenous events is not evident in the training data. Specifically, these events are not tagged with *endogenous* or *exogenous* labels and therefore, it is not possible to construct the set of exogenous events $\mathcal{S}$, well in advance. Existing works [21, 12, 84, 77] tackle this problem by maximizing the likelihood corresponding to the superposition of exogenous and endogenous MTPPs. In such case, the intensity function becomes $\lambda(t) = \mu + \lambda_{\boldsymbol{\theta}}(t \,|\, \mathcal{H}_t)$ and the mark distribution incorporates the effect of exogenous events using additive offsets.

Maximizing the likelihood associated with the superposition of two MTPPs, often turns out to be suboptimal as shown in our experiments (Section 6). This is because, such an approach does not aim to find out the set of exogenous events $\mathcal{S}$ which maximizes likelihood function. In fact, due to the absence of any informed choice about $\mathcal{S}$, such a likelihood function assumes that all the partitions between exogenous and endogenous events $\{(\mathcal{S}, \mathcal{H}_T \backslash \mathcal{S})\}$ are equally likely (Appendix B.1). Such an assumption may not be true in general. We ameliorate this limitation by selecting partition $\{(\mathcal{S}, \mathcal{H}_T \backslash \mathcal{S})\}$ along with the model parameters, which induces the highest value of the likelihood, across all possible partitions and the hypothesis class.

## 3.3 Problem statement

We aim to select the exogenous events from a given stream of unlabeled events and simultaneously train these MTPP models (3) on the remaining events which are endogenous in nature.

**Selecting exogenous events in conjunction with parameter estimation.** Given a stream of events $\mathcal{H}_T$ recorded in the time interval $[0, T)$, we aim to pick $\mathcal{S} \subseteq \mathcal{H}_T$— the candidate set of $k$ exogenous events, *i.e.*, $|\mathcal{S}| \le k$— and concurrently, train the parameters $\boldsymbol{\theta}$ on the set of the endogenous events $\mathcal{H}_T \backslash \mathcal{S}$. Using the intensity functions and the mark distributions defined in Eq. (3)— (4), our goal is to solve the following regularized log-likelihood maximization problem:

$$\underset{\boldsymbol{\theta} \in \Theta, \mathcal{S} \subseteq \mathcal{H}_T}{\text{maximize}} \quad \mathcal{L}(\boldsymbol{\theta}; \mathcal{S}, \mathcal{H}_T), \qquad \text{subject to, } |\mathcal{S}| \le k, \tag{5}$$

$$\text{with, } \mathcal{L}(\boldsymbol{\theta}; \mathcal{S}, \mathcal{H}_T) = \sum_{e_i \in \mathcal{H}_T \backslash \mathcal{S}} \left[ -\rho \|\boldsymbol{\theta}\|_2^2 + \log(\lambda_{\boldsymbol{\theta}}(t_i \,|\, \mathcal{H}_{t_i})) + \log m_{\boldsymbol{\theta}}(y_i \,|\, \mathcal{H}_{t_i}) \right] - \int_0^T \lambda_{\boldsymbol{\theta}}(\tau \,|\, \mathcal{H}_\tau) \, d\tau$$

$$+ |\mathcal{S}| \mathbb{E}[\log \mu] - T \mathbb{E}[\mu] + |\mathcal{S}| \mathbb{E}[\log q]. \tag{6}$$

Here, $\rho$ is the coefficient of the $L_2$ regularizer, $k$ is a pre-specified integer with $k \leq |\mathcal{H}_T|$. The above problem poses a subset selection task in conjunction with likelihood maximization. Parameter estimation along with subset selection has also been addressed in the context of human assisted learning [15, 14], efficient learning [18, 38].

Next, we derive an equivalent representation of our optimization problem (6) as an instance of cardinality constrained set function maximization problem.

**Reformulating** (6) **as a set function optimization problem.** Given a fixed set of the exogenous events $\mathcal{S}$, the optimal value of the model parameters $\boldsymbol{\theta}^*(\mathcal{H}_T \backslash \mathcal{S})$ becomes dependent only on the endogenous events $\mathcal{H}_T \backslash \mathcal{S}$. Therefore, if we define:

$$F(\mathcal{S}) = \mathcal{L}(\boldsymbol{\theta}^*(\mathcal{H}_T \backslash \mathcal{S}); \mathcal{S}, \mathcal{H}_T) \tag{7}$$

then the optimization problem (6) becomes equivalent to solving the following optimization problem.

$$\underset{\mathcal{S} \subseteq \mathcal{H}_T}{\text{maximize}}\, F(\mathcal{S}), \quad \text{subject to, } |\mathcal{S}| \leq k \tag{8}$$

**Hardness analysis.** Given the optimal set of exogenous events $\mathcal{S}^*$, one can easily compute the optimal parameters $\boldsymbol{\theta}^*(\mathcal{H}_T \backslash \mathcal{S}^*)$ for the problem (5) in polynomial time, if the objective (6) is concave with respect to $\boldsymbol{\theta}$ for any fixed $\mathcal{S}$. However, in general, both $\mathcal{S}^*$ and $\boldsymbol{\theta}^*(\mathcal{H}_T \backslash \mathcal{S}^*)$ can not be solved in polynomial time. We formally state this result in the following proposition (Proven in Appendix B.2).

**Proposition 1** *Solving the optimization problem in Eq.* (5) *is NP-hard.*

## 4 Our proposed framework for maximizing $F(\mathcal{S})$

In this section, we present our framework to maximize $F(\mathcal{S})$. More specifically, we first show that the set function $F(\mathcal{S})$ defined in Eq. (7) is monotone and $\alpha$-submodular for a wide variety of MTPPs. Then, we present TPP-SELECT, an approximation algorithm based on a scalable stochastic greedy algorithm [56], which identifies the optimal set of exogenous events from a set of unlabeled events, for any given MTPP model. Finally, we show that such an algorithm enjoys an approximation guarantee even when the learning algorithm provides an imperfect estimate of the trainable parameter $\boldsymbol{\theta}$.

### 4.1 Characterization of $F(\mathcal{S})$

First, we show that $F(\mathcal{S})$ enjoys several useful properties for a wide variet of MTPPs, which allows us to leverage a stochastic greedy algorithm that offers approximation guarantee. Before formally stating those properties, we first present the formal definitions of monotonicity and $\alpha$-submodularity.

**Definition 2** *A function $F(\mathcal{S})$ is monotone non-decreasing if $F(\mathcal{S} \cup \{e\}) - F(\mathcal{S}) \geq 0$ for all $e \in \mathcal{H}_T \backslash \mathcal{S}$. $F(\mathcal{S})$ is $\alpha$-submodular if $F(\mathcal{S} \cup \{e\}) - F(\mathcal{S}) \geq \alpha(F(\mathcal{T} \cup \{e\}) - F(\mathcal{T}))$ for all $e \in \mathcal{H}_T \backslash \mathcal{T}$ and $\mathcal{S} \subseteq \mathcal{T} \subset \mathcal{H}_T$ [28, 86, 20].*

**Monotonicity and $\alpha$-submodularity of $F(\mathcal{S})$ with linear MTPPs.** First, we consider a linear marked temporal point process which is built upon Hawkes process [12, 77, 66]. Here, we model the intensity function $\lambda_{\boldsymbol{\theta}}(t \,|\, \mathcal{H}_t)$ and the mark distribution $m_{\boldsymbol{\theta}}(y \,|\, \mathcal{H}_t)$ for the endogenous events as a linear combination of multiple triggering kernels $\{\kappa_{\bullet}(t, t_i) > 0\}$, *i.e.*,

$$\lambda_{\boldsymbol{\theta}}(t \,|\, \mathcal{H}_t) = \boldsymbol{\theta}_{\lambda}^{\top} \boldsymbol{\kappa}_{\lambda}(t), \qquad m_{\boldsymbol{\theta}}(y \,|\, \mathcal{H}_t) = \text{logistic}(y \cdot \boldsymbol{\theta}_m^{\top} \boldsymbol{\kappa}_m(t)) \tag{9}$$

where $\boldsymbol{\theta}_{\bullet}^{\top} \boldsymbol{\kappa}_{\bullet}(t) = \sum_{v=1}^{d} \theta_{\bullet,v} \sum_{e_i \in \mathcal{H}_t} \kappa_{\bullet,v}(t, t_i)$. Here, $d$ is the dimension of $\boldsymbol{\kappa}$ and $\boldsymbol{\theta} = \{\boldsymbol{\theta}_{\lambda,v}, \boldsymbol{\theta}_{m,v} \,|\, v \in [d]\}$ is the set of trainable parameters. In general, $\kappa_{\bullet,v}(\cdot, \cdot)$ captures different forms of dependencies. For example, in the particular case of information diffusion in social networks, $\lambda_{\boldsymbol{\theta}}$ may correspond the rate of message of a user $u$ with $d$ neighbors and, $\kappa_{\bullet,v}(t, t_i) = e^{-\omega_{\bullet}(t-t_i)} \cdot \mathbf{1}(t_i \text{ is posted by user } v)$ which captures the scenario the effect of an previously occurred event $e_i$ decays over time with the decay factor $\omega_{\bullet} > 0$. In the subsequent theorems, we formally state that $F(\mathcal{S})$ is a monotone and submodular set function (Proven in Appendix C.1).

**Theorem 3 (Monotonicity of linear MTPP)** *Assume that the endogenous events follow a linear MTPP* (9). *Then the set function $F(\mathcal{S})$ defined in Eq.* (7) *is monotone in $\mathcal{S}$ if $\rho \geq \rho_{\min} = (0.5/e) \cdot \kappa_{\max,\lambda}^2 \cdot \exp\left(-2\mathbb{E}[\log(\mu \cdot q)]\right)$, where $\mu$ and $q$ are the intensity and the mark distribution for the exogenous events and $\kappa_{\max,\lambda} = \max_{t \leq T} \|\boldsymbol{\kappa}_{\lambda}(t)\|_2^2$.*

We would like to hightlight that for several linear MTPP models, such an assumption on the regularization parameter $\rho$ is quite loose. For examples, in the previously discussed example of information diffusion with exponential kernels, if we have $\kappa_{\max,\lambda} = 1$ and an exogenous event distribution $\mu, q \sim \text{Gamma}(1,1)$, we have $\rho_{\min} < 0.1$.

**Theorem 4 ($\alpha$-submodularity of linear MTPP)** *Given that the same conditions in Theorem 3 as well as* $\left\| \int_0^\infty \boldsymbol{\kappa}_\lambda(t)dt \right\|_2 \le K < \infty$, $\kappa_{\max,\bullet} = \max_{t \le T} \|\boldsymbol{\kappa}_\bullet(t)\|_2$, $\kappa_{\min,\bullet} = \min_{v \in [d], t \le T} |\kappa_{\bullet,v}(t)|$. *If $K, \kappa_{\max,\bullet}, \kappa_{\min,\bullet} > 0$, then the set function $F(\mathcal{S})$ is $\alpha$-submodular in $\mathcal{S}$ with $\alpha \ge \alpha_F$, where*

$$\alpha_F = \frac{\log\left[\sqrt{2e\rho/\kappa_{\max,\lambda}^2}\right] + \mathbb{E}[\log(\mu \cdot q)]}{\frac{\kappa_{\max,\lambda}}{2\kappa_{\min,\lambda}} + \log\frac{4\kappa_{\max,\lambda}}{\kappa_{\min,\lambda}^2} + \log\left(\sqrt{\frac{2\rho\kappa_{\max,\lambda}}{\kappa_{\min,\lambda}}} + K\right) + \frac{\kappa_{\max,m}^2 \log 2}{2\rho} + \sqrt{\frac{\kappa_{\max,m}^2 \log 2}{\rho}} + \mathbb{E}[\log(\mu \cdot q)]}$$

In the previously discussed example of linear MTPP, which has an exponential kernel having decay factor $\omega$, we have $\alpha_F^* = O(\log\rho/(\log\rho + be^{\omega T}))$, where $b$ is a data dependent constant. Hence, by choosing a low decay factor $\omega$, one can obtain a high value of $\alpha_F$.

*Proof sketches of Theorem 3 and 4:* We prove Theorem 3 in two steps. In the first step, we show that $F(\mathcal{S} \cup \{e\}) - F(\mathcal{S}) \ge \min_{\boldsymbol{\theta}_\lambda}[\rho\|\boldsymbol{\theta}_\lambda\|_2^2 - \log(\boldsymbol{\theta}_\lambda^\top \boldsymbol{\kappa}_\lambda(t))] + \exp\left(-2 \cdot \mathbb{E}[\log(\mu \cdot q)]\right)$. In the second step, we show that this lower bound is greater than $\mathbb{E}[\log\mu + \log q] + \log\left[\sqrt{2e\rho/\kappa_{\max}^2}\right]$ which proves the theorem. To prove Theorem 4, we aim to bound $[F(\mathcal{S} \cup \{e\}) - F(\mathcal{S})]/[F(\mathcal{T} \cup \{e\}) - F(\mathcal{T})]$. To bound the numerator, we use the result of Theorem 3. To bound the denominator, we first show that $F(\mathcal{T} \cup \{e\}) - F(\mathcal{T}) \le \rho\|\boldsymbol{\theta}^*(\mathcal{H}_T \backslash (\mathcal{T} \cup \{e\}))\|_2^2 - \log(\boldsymbol{\theta}_\lambda^*(\mathcal{H}_T \backslash (\mathcal{T} \cup \{e\}))^\top \boldsymbol{\kappa}_\lambda(t)) + \log[1 + \exp\left(-y \cdot \boldsymbol{\theta}_m^*(\mathcal{H}_T \backslash (\mathcal{T} \cup \{e\}))^\top \boldsymbol{\kappa}_m(t)\right)]$. These individual terms are finally bounded using minimum and maximum possible values of $\|\boldsymbol{\theta}\|_2$.

**Monotonicity and $\alpha$-submodularity of Nonlinear MTTPs.** Next, we consider nonlinear MTPPs that do not have any fixed parameterized form, which are trained using deep learning methods [17, 87, 89]. Such models use a sequence encoder— either a recurrent neural network [17] or a transformer [87, 89]— to first derive an embedding $\boldsymbol{h}_i$ of the history of events $\mathcal{H}_{t_i}$ happened until time $t_i$ and then use $\boldsymbol{h}_i$ to capture the generative process of the next event. Characterization of $F(\mathcal{S})$ for such a general MTPP is an extremely challenging task. Therefore, we focus on a class of nonlinear MTPPs which satisfy the following conditions on the intensity function and the mark distribution, *i.e.*,

$$\lambda_{\min}e^{-a\|\boldsymbol{\theta}\|_2} \le \lambda_{\boldsymbol{\theta}}(t \mid \mathcal{H}_t) \le \lambda_{\max}e^{a\|\boldsymbol{\theta}\|_2}, \qquad m_{\min}e^{-b\|\boldsymbol{\theta}\|_2} \le m_{\boldsymbol{\theta}}(\cdot \mid \mathcal{H}_t) \le 1. \qquad (10)$$

Here, $\lambda_{\max}, \lambda_{\min}, m_{\min}$ are constants such that: $\lambda_{\boldsymbol{\theta}=\mathbf{0}}(t \mid \mathcal{H}_t) \in [\lambda_{\max}, \lambda_{\min}]$. We would like to highlight that many popular MTPP processes including recurrent marked temporal process (RMTPP) [17], self-attentive point process [87], transformer Hawkes process [89], neural Hawkes Process [52], etc. satisfy the above conditions. In fact, such an exponential bound is tight for the intensity function used in RMTPP [17], since its intensity function uses an exponential activation function in the last layer. In Appendix C.3, we discuss the representations of different MTPPs in the form defined in Eq. (10). Next, we show the monotonicity and submodularity of $F(\mathcal{S})$ in the following theorems (Proven in Appendix C.2)

**Theorem 5 (Monotonicity of nonlinear MTPP)** *Assume that the endogenous events follow a nonlinear MTPP with exponentially bounded intensity function, as described in Eq. (10) with $\lambda_{\max} \le \exp(-\mathbb{E}[\log(\mu \cdot q)])$. Then the set function $F(\mathcal{S})$ defined in Eq. (7) is monotone in $\mathcal{S}$ if $\rho \ge 0.25\, a^2/\left(\log[1/\lambda_{\max}] + \mathbb{E}[\log(\mu \cdot q)]\right)$, where $\mu$ and $q$ are the intensity and the mark distribution for the exogenous events.*

**Theorem 6 ($\alpha$-submodularity of nonlinear MTPP)** *Given the same conditions in Theorem 5 and $\int_0^\infty \|\lambda_{\mathbf{0}}(t \mid \mathcal{H}_t)\|_2 dt \le \Lambda_{\max} < \infty$, the set function $F(\mathcal{S})$ is $\alpha$-submodular in $\mathcal{S}$ with $\alpha \ge \alpha_F$, where $\alpha_F$ is equal to*

$$\frac{\log\left[\exp\left(-a^2\mathbb{E}[\log(\mu \cdot q)]/4\right)/\lambda_{\max}\right]}{\Lambda_{\max} + (a+b)\sqrt{\frac{\log(1/\lambda_{\min}m_{\min}) + \Lambda_{\max}}{\rho}} + 2\log(1/\lambda_{\min}m_{\min}) + \mathbb{E}[\log(\mu \cdot q)]} \qquad (11)$$

We would like to point out that $\lambda_{\boldsymbol{\theta}}(t \mid \mathcal{H}_T) = \lambda_0 > 0$ for the nonlinear MTPPs having exponential bounds. Indeed, in practice, we encounter several existing models [17, 89, 87] that choose activation functions, *e.g.*, Softplus, exponential function, having nonzero intensity functions even when the

parameters $\boldsymbol{\theta} = \mathbf{0}$. However, for the linear MTPPs described in Eq. (9), $\lambda_{\boldsymbol{\theta}}(t \,|\, \mathcal{H}_t) = 0$ for $\boldsymbol{\theta} = \mathbf{0}$. Therefore, the set of nonlinear MTPPs considered above does not cover the class of linear MTPPs (9) and therefore, the results and the proofs in Theorems 5-6 are different from Theorems 3-4. Appendix D characterizes other types of nonlinear MTPPs.

## 4.2 TPP-SELECT: Approximation algorithm for maximizing $F(\mathcal{S})$

Finally, we present TPP-SELECT, an approximation algorithm based on a stochastic greedy algorithm proposed in [56] in order to solve the optimization problem defined in (8).

**Outline of TPP-SELECT.** In Algorithm 1, we summarize TPP-SELECT to maximize $F(\mathcal{S})$. It is a randomized iterative algorithm based on the stochastic greedy algorithm proposed in [56], which keeps picking up one element per each of $k$ iterations (refer to the for loop in line no. 2). In each iteration, given the current estimate of $\mathcal{S}$, Algorithm 1 picks up a candidate exogenous event $v$ which maximizes $F(\mathcal{S} \cup \{v\})$. However, in contrast to the well known (deterministic) greedy algorithm [61], stochastic greedy algorithm computes the maximum only on a subset of elements— sampled from the current candidate set of elements.

**Advantage over deterministic greedy algorithm.** The stochastic greedy algorithm is the randomized version of well known greedy algorithm [61]. If we put $\mathcal{V} = \mathcal{H}_T \backslash \mathcal{S}$, then the

---

**Algorithm 1** Stochastic greedy algorithm [56]

**Require:** The set of observed events $\mathcal{H}_T$, the final time $T$, the regularization parameter $\rho$, the number of exogenous events $k$.
1: $\mathcal{S} \leftarrow \emptyset$
2: **for** $i \in [k]$ **do**
3:     Randomly draw a subset $\mathcal{V}$ from $\mathcal{H}_T \backslash \mathcal{S}$
4:     **for** $v \in \mathcal{V}$ **do**
5:         $\boldsymbol{\theta} \leftarrow \text{TRAIN}(\mathcal{L}; \mathcal{S} \cup \{v\})$
6:         $F(\mathcal{S} \cup \{v\}) \leftarrow \mathcal{L}(\boldsymbol{\theta}; \mathcal{S}, \mathcal{H}_T)$
7:         $\widehat{\boldsymbol{\theta}}(\mathcal{H}_T \backslash (\mathcal{S} \cup v)) = \boldsymbol{\theta}$
8:     **end for**
9:     $v^* \leftarrow \text{argmax}_{v \in \mathcal{V}} F(\mathcal{S} \cup \{v\})$
10:     $\mathcal{S} \leftarrow \mathcal{S} \cup \{v^*\}$
11: **end for**
12: **Return** $\mathcal{S}, \widehat{\boldsymbol{\theta}}(\mathcal{H}_T \backslash (\mathcal{S} \cup v^*))$

---

stochastic greedy algorithm becomes equivalent to the greedy algorithm. However, we need to train the model $|\mathcal{V}|$ times (due to the for loop in lines 4–8) per each iteration. As a result, the deterministic greedy algorithm would have been extremely inefficient, as it needs to greedily search for the optimal event over all the possible candidate events $\mathcal{V}$. Therefore, reducing the search space to a subset $\mathcal{V}$ would significantly improve the efficiency.

**Approximation guarantees.** Next, we provide the approximation guarantee of Algorithm 1 for solving the optimization problem defined in (8). Existing works [56, 37], provide approximation guarantees of Algorithm 1 when the exact value of $F(\mathcal{S})$ is available. However, in the context of our problem, a training algorithm may not provide the optimal value of model parameters $\boldsymbol{\theta}^*(\mathcal{H}_T \backslash \mathcal{S})$, even when the MTPP model is linear. Hence, we mostly obtain an imperfect estimate of the model parameters $\boldsymbol{\theta}^*(\mathcal{H}_T \backslash \mathcal{S})$. To this end, we derive the approximation guarantee of Algorithm 1 when the underlying training algorithm provides an imperfect estimate of the model parameters $\boldsymbol{\theta}^*(\mathcal{H}_T \backslash \mathcal{S})$.

We would like to highlight that the following approximation guarantees apply to the cardinality constrained maximization problem of any monotone $\alpha$-submodular function and therefore is of independent interest.

**Theorem 7** *Assume that the training algorithm* TRAIN() *in Algorithm 1 provides imperfect estimates of the underlying model parameters, with* $\mathcal{L}(\boldsymbol{\theta}^*(\mathcal{H}_T \backslash \mathcal{S}); \mathcal{S}; \mathcal{H}_T) - \mathcal{L}(\widehat{\boldsymbol{\theta}}(\mathcal{H}_T \backslash \mathcal{S}); \mathcal{S}; \mathcal{H}_T)) \leq \epsilon$ *for all* $\mathcal{S}$ *and that it runs with* $|\mathcal{V}| = O\left(\frac{|\mathcal{H}_T|}{k} \log(1/\delta)\right)$ *(cf. line number 3). Then,*

$$\mathbb{E}[F(\mathcal{S})] \geq (1 - \exp\left(-\alpha_F^*\right) - \alpha_F^* \cdot \delta) \cdot F(\mathcal{S}^*) - k\epsilon \tag{12}$$

*where the expectation is taken over many draws of* $\mathcal{V}$ *(cf. line number 3),* $\mathcal{S}^*$ *is the solution of the optimization problem* (8) *and* $\alpha_F^*$ *is the submodularity ratio, computed using Theorems 4 and 6.*

The above approximation guarantee in the presence of imperfect parameter estimates is related to several existing works [20, 63, 31, 30, 32, 72]. However, they do not focus on cardinality constrained maximization of $\alpha$-submodular functions in general and not the well known stochastic greedy algorithm in particular. Note that, when we have a perfect training algorithm, *i.e.*, $\epsilon = 0$, Theorem 7 reduces to similar results in [56, 37].

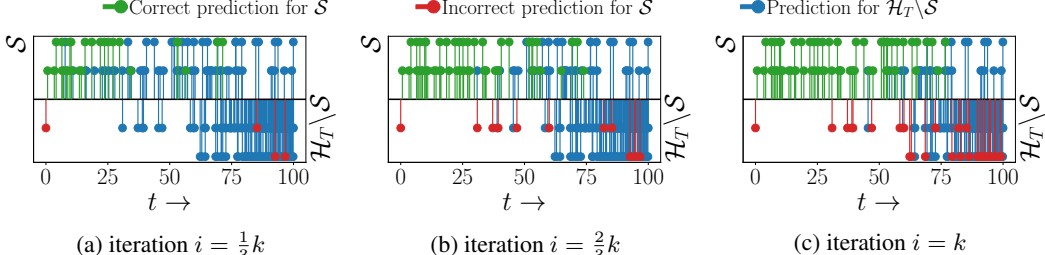

(a) iteration $i = \frac{1}{3}k$     (b) iteration $i = \frac{2}{3}k$     (c) iteration $i = k$

Figure 1: Snapshots of output of Algorithm 1 during one third, two third and final stages of executions. The exogenous intensity $\mu$ is tuned in such a way that $|\mathcal{S}| = 0.3\,|\mathcal{H}_T|$. Moreover, we also set $k = 0.3\,|\mathcal{H}_T|$. Top half (bottom half) of each figure indicates ground truth exogenous events $\mathcal{S}$ (endogenous events $\mathcal{H}_T \backslash \mathcal{S}$). Blue, green and red events indicate predicted endogenous events, correctly predicted exogenous events and wrongly predicted endogenous events. We observe that, the algorithm is able to predict the exogenous events with significant accuracy, as the number of wrongly predicted exogenous events is significantly lower than the correctly predicted exogenous events. The classification accuracy of final stage is 0.806.

## 5 Experiments with synthetic data

In this section, we create synthetic examples containing endogenous or exogenous label and evaluate TPP-SELECT from both qualitative and quantitative perspectives.

### 5.1 Experimental setup

**Data generation.** At the very outset, we generate data using linear MTPP defined in Eq. (9). More specifically, we set the number of marks $|\mathcal{C}| = 2$. In order to sample the endogenous events, we set $d = 2$ and draw unidimensional parameters $\theta_{\lambda,v}, \theta_{m,v} \sim \text{Unif}[0,1]$ for $v \in \{1,2\}$. We use the triggering kernel $\kappa_{\lambda,v}(t, t_i) = \exp(-\omega_\lambda(t - t_i))$ and $\kappa_{m,v} = \exp(-\omega_m(t - t_i))$, where $\omega_\lambda, \omega_m \in \text{Unif}[0,1]$ for $v \in [d]$. For exogenous messages, we sample $\mu$ and $q$ using the exogenous distribution (4) with $\alpha_\lambda, \beta_\lambda, \alpha_m, \beta_m = 1$. Finally, we use Ogata's thinning algorithm [62] to generate both endogenous and exogenous events.

**Baselines.** We compare our method against several data selection based unsupervised robust learning methods: (i) EM algorithm [70] which first assigns a constant probability $p$ which models the likelihood of an event $e$ being exogenous and then estimates it along with model parameters $\boldsymbol{\theta}$ using expectation maximization; (ii) $K$-means [70] which uses $K$-means clustering to cluster the events in $K = 2$ groups and then label the smallest size cluster as $\mathcal{S}$, the exogenous event set; (iii) Principal Component Analysis (PCA) [71] which first finds a low-dimensional projection of the representation vector of the events, then sorts them in the decreasing order of the reconstruction error and finally, choose top-$k$ events as the exogenous event set $\mathcal{S}$; and, (iv) Facility-Location [1] which maximizes a facility location function to select the set of exogenous events $\mathcal{S}$. For K-means, PCA, we directly used implementation available in Keras [8]. For Facility location, we used apricot library [68].

### 5.2 Results

We qualitatively show how TPP-SELECT keeps selecting the exogenous events throughout the progression of Algorithm 1. To that aim, we tune the exogenous intensity $\mu$ in such a way that $|\mathcal{S}| = 0.3\,|\mathcal{H}_T|$ and run Algorithm 1 with $k = 0.3\,|\mathcal{H}_T|$. Figure 1 summarizes the results,

| TPP-Select | EM | K-means | PCA | Fac-loc |
|---|---|---|---|---|
| 0.806 | 0.554 | 0.620 | 0.562 | 0.299 |

Table 1: Accuracy of exogenous event selection in synthetic data in the figure 1 for different methods.

which shows that: (i) the algorithm is able to predict the exogenous events with remarkable accuracy, as the number of wrong predictions is significantly lower than the correct predictions; (ii) the performance of algorithm is strikingly accurate during the initial stage, as the number of wrong predictions of exogenous events is significantly low; and, (iii) even when TPP-SELECT makes wrong predictions about $e \in \mathcal{S}$, there always exists some correct ground truth exogenous event around it which indicates that TPP-SELECT was almost able to predict them correctly.

Next, we compare our method against the baselines. Table 1 summarizes the results, which shows that our method outperforms the baselines.

|            | TPP-SELECT        | EM                | PCA               | Fac-Loc           |
| ---------- | ----------------- | ----------------- | ----------------- | ----------------- |
| Club       | **0.32 ±0.02**    | 0.18 ±0.02        | 0.28 ±0.01 | 0.25 ±0.01        |
| Election   | **0.58 ±0.03**    | 0.31 ±0.01        | 0.19 ±0.00        | 0.44 ±0.03 |
| Series     | 0.18 ±0.02        | **0.27 ±0.02**    | 0.26 ±0.01 | 0.16 ±0.01        |
| Verdict    | **0.33 ±0.04**    | 0.27 ±0.01 | 0.18 ±0.00        | 0.10 ±0.01        |
| BookOrder  | **0.96 ±0.01**    | 0.41 ±0.03        | 0.92 ±0.01 | 0.46 ±0.01        |

Table 2: Accuracy of exogenous event selection of TPP-SELECT, EM [70], PCA [71] and Facility-Location [1] across all datasets, when we inject $|\mathcal{S}| = 0.4|\mathcal{H}_T|$ exogenous events. Numbers in bold font (underline) indicate the best (second best) performer. We observe that in a majority of the cases, TPP-SELECT outperforms the baselines.

# 6 Experiments with real data

In this section, we provide a comprehensive evaluation of our proposal on several real world datasets, which shows that our method outperforms several data selection based robust learning methods, as well as the underlying base MTPP model which is trained over all observations including exogenous events. Appendix G contains additional experiments.

## 6.1 Experimental setup

**Datasets.** We considered five real world datasets, also summarized in Appendix F, which are (i) Club, (ii) Election, (iii) Series, (iv) Verdict and (v) BookOrder. Among these datasets, the first four datasets were gathered from the works [12, 84, 41], whereas, the last dataset were gathered from [17] [1].

**Baselines.** We compare our method against the four data selection based unsupervised robust learning methods, presented in Section 5.1.

## 6.2 Results

**Comparison with baselines.** We first compare the performance of TPP-SELECT against the baselines in terms of the classification accuracy for exogenous event selection. To that aim, we inject 40% exogenous events and present to the event selection methods for identifying these events. Here, we consider recurrent marked temporal point process (RMTPP) [17] as our base MTPP model $(\lambda_{\boldsymbol{\theta}}, m_{\boldsymbol{\theta}})$. Note that, unlike synthetic dataset, K-means is not able to predict any exogenous events correctly. We believe that this poor performance is due to the complexity of the underlying generative process in real datasets, which renders the feature distance between events to be the poor predictors of exogenous labels. Table 2 summarizes the results of our method and the other baselines, which shows that (i) our method consistently outperforms the baselines in majority of the cases, as it aims to select the events in conjunction with the model parameters, whereas the baselines except for EM algorithm selects the events in a model-agnostic manner; (ii) except for Series and Verdict datasets, EM algorithm fares poorly across all other datasets, which we believe is due to the time-invariant modeling of probability of the exogenous events; and (iii) PCA is the second best winner in majority of the datasets.

**Comparison with the base MTPP model.** Next, we investigate how well TPP-SELECT performs in comparison to the underlying base MTPP which models the event dynamics using the superposition between endogenous and exogenous MTPPs and thus, is trained on the entire sequence of events including the exogenous events. Given a sequence of observed events, we split it into 80% training and 20% test sets. We use TPP-SELECT on the training set $\mathcal{H}_T$, where it simultaneously selects the exogenous events $\mathcal{S}$ and trains the model parameters $(\lambda_{\boldsymbol{\theta}}, m_{\boldsymbol{\theta}})$ on the endogenous events $\mathcal{H}_T \backslash \mathcal{S}$. Once we train the model, we use Ogata's thinning algorithm [62] to simulate future events to predict the unseen events in the test set $\mathcal{T}$. To this end, we evaluate the performance on in terms of mean absolute error (MAE), *i.e.*, $\frac{1}{|\mathcal{T}|} \sum_{e=(t,y) \in \mathcal{T}} \mathbb{E}[|t - \hat{t}|]$ and mark prediction error (MPE), *i.e.*, $\frac{1}{|\mathcal{T}|} \sum_{e=(t,y) \in \mathcal{T}} \Pr(y \neq \hat{y})$. Here, MAE and MPE measure the errors in predicting the arrival times and the marks of the future events.

Table 3 summarizes the results, which shows that TPP-SELECT performs better than the underlying base MTPP model across all datasets. Such observations imply that a model aware exogenous event filtering improves the predictive performance than learning a superposition of two models.

---

[1]The datasets are public and anonymized. We collect them from https://github.com/paramita1024/demarcation/tree/master/data and https://github.com/musically-ut/tf_rmtpp/tree/master/data/real. They are under MIT License. No dataset contains personally identifiable information or offensive content.

| | Time prediction error in terms of MAE $\mathbb{E}(|t - \widehat{t}|)$ | | | | Mark prediction error $\Pr(y \neq \widehat{y})$ | | | |
|---|---|---|---|---|---|---|---|---|
| | Base model: RMTPP | | Base model: THP | | Base model: RMTPP | | Base model: THP | |
| | TPP-SELECT | Base MTPP | TPP-SELECT | Base MTPP | TPP-SELECT | Base MTPP | TPP-SELECT | Base MTPP |
| Club | **3.57±0.15** | 3.67±0.05 | **11.51±0.07** | 11.93±0.08 | 0.60±0.11 | **0.59±0.05** | **0.36±0.01** | 0.42±0.01 |
| Election | **8.50±0.02** | 8.50±0.01 | **30.79±0.02** | 30.99±0.03 | **0.43±0.02** | 0.67±0.03 | **0.38±0.00** | 0.41±0.01 |
| Series | **1.15±0.05** | 1.71±0.07 | **9.12±0.17** | 9.99±0.20 | **0.59±0.00** | 0.69±0.01 | **0.57±0.00** | 0.58±0.00 |
| Verdict | **2.27±0.00** | 2.27±0.00 | **7.88±0.01** | 8.11±0.05 | **0.57±0.00** | 0.71±0.01 | **0.57±0.00** | 0.60±0.01 |
| BookOrder | **0.07±0.00** | 0.07±0.00 | **0.09±0.00** | 0.11±0.01 | **0.45±0.00** | 0.49±0.00 | **0.37±0.01** | 0.44±0.00 |

Table 3: Performance in terms of $\mathbb{E}(|t - \widehat{t}|)$, *i.e.*, the mean absolute error (MAE) for time prediction (left half) and the misclassification error $\Pr(y \neq \widehat{y})$ for the mark prediction (right half) for TPP-SELECT and the corresponding base MTPP model which is the superposition of endogenous and exogenous MTPP models. Here, this base MTPP model is learned over the entire sequence of observed events. In all experiments, we considered 80% training and 20% test set.

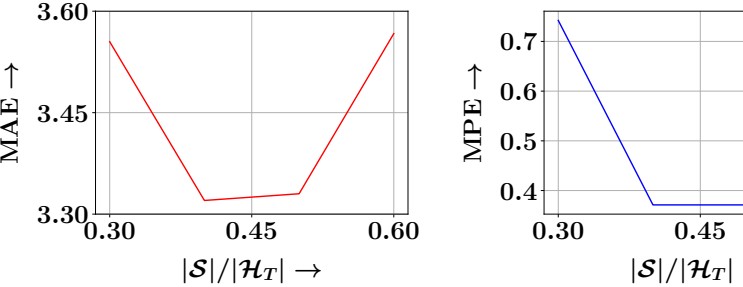

Figure 2: Variation of mean absolute error $\mathbb{E}(|t - \widehat{t}|)$ and mark prediction error (MPE) $\Pr(y \neq \hat{y})$ against $k$, the maximum number of exogenous events, on Club dataset. We observe that in both the cases when $k$ is too small or high, then TPP-SELECT performs poorly. However, there exists a sweet spot of $k$, where TPP-SELECT shows its best performance.

**Effect of $k$.** Finally, we investigate how $k$, the pre-specified number of exogenous events, impacts the performance of TPP-SELECT. Figure 2 summarizes the results, which show that (i) TPP-SELECT performs poorly when $k$ is either too small or too large, and (ii) there exists a sweet spot of $k$, where TPP-SELECT shows the best peformance.

## 7 Conclusion

In this paper, we aim to identify the set of exogenous events from a set of unlabeled events for learning marked temporal point process. To that goal, we have developed TPP-SELECT, a novel framework that selects the possible set of exogenous events from a set the observed events, under any given MTPP model. More specifically, we first cast this problem as an instance of exogenous event set selection problem, in conjunction with parameter estimation problem which is NP-hard, in general. However, we show that solving such a problem is equivalent to maximizing a monotone and $\alpha$-submodular set function which leads us to develop TPP-SELECT— an approximation algorithm built upon a stochastic greedy algorithm which was originally proposed in [64]. In this work, we show that it also enjoys an approximation guarantee for $\alpha$-submodular set function maximization, even when the learning algorithm provides an imperfect estimate of the trained parameters. Our experiments show that TPP-SELECT is able to outperform a wide variety of baselines based on data-selection from robust learning literature.

Our work opens several venues of future work. For example, our method can be easily applied to identify external influence in Twitter— such a situation can immediately beneficial in a wide variety of applications that include viral marketing, misinformation detection, etc. However, by deploying our method as-it-is can lead to privacy leakage of users. To counter this effect, one can design a differentially private framework to mitigate this effect, by building upon the work by Mitrovic et al. [59]. In our work, we do not consider a multivariate point process. It would be interesting to develop exogenous event selection method for such processes. Finally, it would be valuable to extend our method to an online setting, where the task is to select the set of exogenous events in the face of an online stream of events.

**Acknowledgments and Disclosure of Funding**

The project was partially funded by Google Research Grant and DST Inspire Faculty Grant. This work is also supported by the National Science Foundation under Grant No. IIS-2106937. Any opinions, findings, and conclusions or recommendations expressed in this material are those of the author(s) and do not necessarily reflect the views of the National Science Foundation, Google or DST.

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
