# Supplementary Material
# (Learning to Select Exogenous Events for Marked Temporal Point Process)

## Table of Contents

# A Broader impact

## A.1 Positive potential impacts

At the very outset, our problem aims at selecting exogenous events from a stream of events. Such a problem has a wide variety of applications in real life,

**Preventing misinformation flow.** Misinformation is easily spread in social networks, which often has a negative impact on people's social life. For such a information cascade like fake news/rumor, our algorithm may be useful.

**Viral marketing.** While making marketing strategy, we can use our approach to get to know which user is more sensitive to exogenous events. These users can be used as seed users for improvement in viral marketing.

**Improvement in cyber security.** Cyber attacks cause tremendous damage to institutes and individuals. For example, DDoS attack sends a very large number of malicious requests in a short period of time with unpredictable pattern. Unlike real requests which are often relevant to historical events, these malicious requests usually shows exogenous pattern as they are history-irrelevant. In the context of cyber security, we can detect potential malicious events and prevent system from attacks using our model.

## A.2 Negative potential impacts

**Privacy concerns.** Our model can select exogenous events from a series of events, and can parametrically estimate relationship between users. Currently, it does not protect privacy of users. In particular, the social relationship of users can be leaked through the parameters of the trained model. Therefore, it would be important to design privacy preserving event selection method using tools from differential privacy.

**Bias introduced in our model**: We have not tried anything to eliminate the unfairness and bias in training and prediction process. As a result, it can introduce bias into the algorithm. For example, in a cascade extracted from a community that is broadly represented by a specific group of persons, our model can label an event triggered by a user from any other group as exogenous. Therefore, it would be an important task to introduce fairness within the algorithm.

## B  Discussion on problem setup

### B.1  Benefits and limitations of our approach

**Benefits of our approach.** Existing approaches [21, 12] model the arrival time dynamics of an event using superposition of exogenous and endogenous intensity functions, *i.e.*, $\mu + \lambda_{\boldsymbol{\theta}}$. We first put such an approach in context. More specifically, we first consider a partition $(\mathcal{S}, \mathcal{H}_T \backslash \mathcal{S})$ for endogenous and exogenous events. Then, we model the density of arrival times $e = (y, t)$ as,

$$p_{\boldsymbol{\theta}}(\mathcal{S}, \mathcal{H}_T \backslash \mathcal{S}) = \mu^{|\mathcal{S}|} \prod_{e \in \mathcal{H}_T \backslash \mathcal{S}} \lambda_{\boldsymbol{\theta}}(t \,|\, \mathcal{H}_t) \exp\left(-\mu T - \int_0^T \lambda_{\boldsymbol{\theta}}(\tau \,|\, \mathcal{H}_\tau) d\tau\right). \tag{13}$$

Now, if the prior probability of partition $(\mathcal{S}, \mathcal{H}_T \backslash \mathcal{S})$ is $p_{\text{prior}}(\mathcal{S})$, then the likelihood of (only) the arrival times of the observed events becomes

$$p_{\boldsymbol{\theta}}(\mathcal{H}_T) = \sum_{\mathcal{S} \in 2^{\mathcal{H}_T}} p_{\boldsymbol{\theta}}(\mathcal{S}, \mathcal{H}_T \backslash \mathcal{S}) \, p_{\text{prior}}(\mathcal{S})$$

$$= \sum_{\mathcal{S} \in 2^{\mathcal{H}_T}} \left[ \prod_{e \in \mathcal{H}_T} \mu^{\mathbb{I}[e \in \mathcal{S}]} [\lambda_{\boldsymbol{\theta}}(t \,|\, \mathcal{H}_t)]^{\mathbb{I}[e \notin \mathcal{S}]} p_{\text{prior}}(\mathcal{S}) \right] \times \exp\left(-\mu T - \int_0^T \lambda_{\boldsymbol{\theta}}(\tau \,|\, \mathcal{H}_\tau) d\tau\right) \tag{14}$$

Now, if $p_{\text{prior}}(\mathcal{S})$ is constant, *i.e.*, $p_{\text{prior}}(\mathcal{S}) = p_{\text{prior}}$, then it immediately gives,

$$p_{\boldsymbol{\theta}}(\mathcal{H}_T) \propto \prod_{e \in \mathcal{H}_T} (\mu + \lambda_{\boldsymbol{\theta}}(t \,|\, \mathcal{H}_t)) \times \exp\left(-\mu T - \int_0^T \lambda_{\boldsymbol{\theta}}(\tau \,|\, \mathcal{H}_\tau) d\tau\right) \tag{15}$$

Hence, such an approach [21, 12] captures a particular case when every partition is equally likely, which may not be true in general. To avoid this problem, we aim to find $\mathcal{S}$ which maximizes the underlying likelihood function.

Furthermore, note that the exogenous distribution parameters $\mu$ and $q$ in existing works [21, 12] are often treated as deterministic variables, which, however, cannot characterize the uncertainty of the exogenous influence. We tackle this challenge by modeling the uncertainty, *i.e.*,

$$\lambda(t) = \mu \sim \text{Gamma}(\alpha_\lambda, \beta_\lambda), \qquad m(y) = q \sim \text{Beta}(\alpha_{m,y}, \beta_{m,y}). \tag{16}$$

**Limitations of our approach.** Apart from the limitations addressed in Appendix A and Conclusion in the main paper, another key technical limitation of our approach is lack of inherent scalability. The stochastic greedy algorithm does reduce the running time by comparing across a limited amount of data. However, such an approach can often affect the predictive performance. It would be interesting to consider similar setup like [39] to improve scalability of our method.

### B.2  Hardness analysis

**Proposition 1** *Solving the optimization problem in Eq. (5) is NP-hard.*

**Proof.** We consider the particular case of MTPP models where the intensity function is constant.

$$\lambda_{\boldsymbol{\theta}}(t \,|\, \mathcal{H}_t) = \lambda_0, \qquad m_{\boldsymbol{\theta}}(y \,|\, \mathcal{H}_t) = m_{\boldsymbol{\theta}}(y \,|\, \mathcal{H}_t). \tag{17}$$

Moreover, we consider the special case when $\mathcal{C} = \{+1, -1, 0\}$ and

$$m_{\boldsymbol{\theta}}(y \,|\, \mathcal{H}_t) = \begin{cases} e^{-(1 - y \cdot \boldsymbol{\theta}^\top \boldsymbol{\phi}(\mathcal{H}_t))^2} & \text{if, } y \in \{+1, -1\} \\ 1 - \sum_{y \in \{-1, +1\}} e^{-(1 - y \cdot \boldsymbol{\theta}^\top \boldsymbol{\phi}(\mathcal{H}_t))^2} & \text{if, } y = 0, \end{cases} \tag{18}$$

$$\lambda_0 = \exp(\mathbb{E}[\log(\mu \cdot q)]). \tag{19}$$

Here $\boldsymbol{\phi}(\mathcal{H}_t)$ is a representation vector for $\mathcal{H}_t$. We assume that (i) $\boldsymbol{\Phi} = [\boldsymbol{\phi}^\top(\mathcal{H}_{t_1}); \boldsymbol{\phi}^\top(\mathcal{H}_{t_N}); ..; \boldsymbol{\phi}^\top(\mathcal{H}_{t_N})]$ has full row rank $N$ and (ii) $\mathcal{H}_T$ only contains marks $y \in \{+1, -1\}$ and (iii) $\rho = 0$. Then the log-likelihood (6) becomes:

$$\sum_{e_i \in \mathcal{H}_T \setminus \mathcal{S}} [\log(\lambda_{\boldsymbol{\theta}}(t_i \,|\, \mathcal{H}_{t_i})) + \log m_{\boldsymbol{\theta}}(y_i \,|\, \mathcal{H}_{t_i})] - \int_0^T \lambda_{\boldsymbol{\theta}}(\tau \,|\, \mathcal{H}_\tau)\, d\tau + |\mathcal{S}| \mathbb{E}[\log(\mu \cdot q)] - T\mathbb{E}[\mu]$$

$$= \sum_{e_i \in \mathcal{H}_T \setminus \mathcal{S}} \log m_{\boldsymbol{\theta}}(y_i \,|\, \mathcal{H}_{t_i}) - T\,\lambda_0 \,+ |\mathcal{S}|\, \mathbb{E}[\log(\mu \cdot q)] + (|\mathcal{H}_T| - |\mathcal{S}|) \log \lambda_0 - T\, \mathbb{E}[\mu]$$

$$\overset{(i)}{=} - \sum_{e_i \in \mathcal{H}_T \setminus \mathcal{S}} (1 - y_i \cdot \boldsymbol{\theta}^\top \phi(\mathcal{H}_{t_i}))^2 - T\,\lambda_0 \,- T\, \mathbb{E}[\mu] \tag{20}$$

The equality (i) is obtained by putting the particular instance described in Eqs. (18) — (19). The next steps of the proof follows directly from De et al. [15, Proof of Theorem 1], which is described below to make the proof self-contained. More specifically, we first define $\boldsymbol{\theta}_0 = \boldsymbol{\Phi}_R^{-1}(\boldsymbol{y} - \boldsymbol{r})$, where $\boldsymbol{y} = [y_1, .., y_N]$, $\boldsymbol{r} \in \mathbb{R}^N$. where $\boldsymbol{\Phi}_R^{-1}$ is the right inverse of $\boldsymbol{\Phi}$ an $\boldsymbol{r}$ is an arbitrary real vector. A trite calculation shows that $\boldsymbol{\theta}_0^\top \phi(\mathcal{H}_{t_i}) = y_i - r_i$. Finally, if we define a new variable $\boldsymbol{\theta}' = \boldsymbol{\theta} - \boldsymbol{\theta}_0$, then we can write as follows:

$$(1 - y_i \cdot \boldsymbol{\theta}^\top \phi(\mathcal{H}_{t_i}))^2 = (1 - y_i \cdot \boldsymbol{\theta}'^\top \phi(\mathcal{H}_{t_i}) - y_i^2 + y_i r_i)^2$$
$$= (r_i - \boldsymbol{\theta}'^\top \phi(\mathcal{H}_{t_i}))^2 \qquad \text{(Since } y_i \in \pm 1) \tag{21}$$

Hence, maximizing the objective defined Eq. (20) w.r.t. $\boldsymbol{\theta}$ and $\mathcal{S}$ is equivalent to minimizing the $\sum_{e_i \in \mathcal{H}_T} (r_i - \boldsymbol{\theta}'^\top \phi(\mathcal{H}_{t_i}))^2$ which is known to be NP-hard [4].

## C  Proofs of the technical results in Section 4

Unless otherwise stated, we denote $\|\cdot\|$ as $\|\cdot\|_2$ and $\boldsymbol{a} \preceq \boldsymbol{b}$ means pointwise inequality, *i.e.*, $a_i \geq b_i$ for all $i \leq \dim(\boldsymbol{a})$

### C.1  Monotonicity and approximate submodularity for linear MTPP

**Theorem 3** *Assume that the endogenous events follow a linear MTPP (9). Then the set function $F(\mathcal{S})$ defined in Eq. (7) is monotone in $\mathcal{S}$ if $\rho \geq \rho_{\min} = (0.5/e) \cdot \kappa_{\max,\lambda}^2 \cdot \exp\big(-2\,\mathbb{E}[\log(\mu \cdot q)]\big)$, where $\mu$ and $q$ are the intensity and the mark distribution for the exogenous events and $\kappa_{\max,\lambda} = \max_{t \leq T} \|\boldsymbol{\kappa}_\lambda(t)\|_2^2$.*

**Proof.** From Lemma 9 in Appendix D, we have that: for $e = (y, t) \notin \mathcal{S}$,

$$F(\mathcal{S} \cup \{e\}) - F(\mathcal{S}) \geq \rho \|\boldsymbol{\theta}^*(\mathcal{H}_T \setminus \mathcal{S})\|^2 - \log \lambda_{\boldsymbol{\theta}^*(\mathcal{H}_T \setminus \mathcal{S})}(t \,|\, \mathcal{H}_t) + \mathbb{E}[\log(\mu \cdot q)]$$

$$\geq \min_{\boldsymbol{\theta}_\lambda} [\rho \|\boldsymbol{\theta}_\lambda\|_2^2 - \log(\boldsymbol{\theta}_\lambda^\top \boldsymbol{\kappa}_\lambda(t))] + \mathbb{E}[\log(\mu \cdot q)] \tag{22}$$

$$\overset{(i)}{\geq} \min_{\boldsymbol{\theta}_\lambda} \left[ \rho \frac{(\boldsymbol{\theta}_\lambda^\top \boldsymbol{\kappa}_\lambda(t))^2}{\|\boldsymbol{\kappa}_\lambda(t)\|^2} - \log(\boldsymbol{\theta}_\lambda^\top \boldsymbol{\kappa}_\lambda(t)) \right] + \mathbb{E}[\log(\mu \cdot q)]$$

$$\geq \min_x \left[ \frac{\rho x^2}{\kappa_{\max,\lambda}^2} - \log(x) \right] + \mathbb{E}[\log(\mu \cdot q)]$$

$$\overset{(ii)}{\geq} \mathbb{E}[\log(\mu \cdot q)] + \log \left[ \sqrt{\frac{2e\rho}{\kappa_{\max,\lambda}^2}} \right]$$

$$> 0, \tag{23}$$

where inequality (i) is due to Cauchy-Schwartz inequality $\|\boldsymbol{a}^\top \boldsymbol{b}\|_2 \leq \|\boldsymbol{a}\|_2 \cdot \|\boldsymbol{b}\|_2$; inequality (ii) is due to Proposition 11 and the last inequality holds true by assumption on $\rho$.

**Theorem 4** *Given that the same conditions in Theorem 3 as well as $\left\| \int_0^\infty \boldsymbol{\kappa}_\lambda(t) dt \right\| \leq K < \infty$, $\kappa_{\max,\bullet} = \max_{t \leq T} \|\boldsymbol{\kappa}_\bullet(t)\|$, $\kappa_{\min,\bullet} = \min_{v \in [d], t \leq T} |\kappa_{\bullet,v}(t)|$. If $K, \kappa_{\max,\bullet}, \kappa_{\min,\bullet} > 0$, then the set*

*function $F(\mathcal{S})$ is $\alpha$-submodular in $\mathcal{S}$ with*

$$\alpha \geq \alpha_F = \frac{\log\left[\sqrt{2e\rho/\kappa_{\max,\lambda}^2}\right] + \mathbb{E}[\log(\mu \cdot q)]}{\frac{\kappa_{\max,\lambda}}{2\kappa_{\min,\lambda}} + \log\frac{4\kappa_{\max,\lambda}}{\kappa_{\min,\lambda}^2} + \log\left(\sqrt{\frac{2\rho\kappa_{\max,\lambda}}{\kappa_{\min,\lambda}}} + K\right) + \frac{\kappa_{\max,m}^2\log 2}{2\rho} + \sqrt{\frac{\kappa_{\max,m}^2\log 2}{\rho}} + \mathbb{E}[\log(\mu \cdot q)]} \tag{24}$$

**Proof.** We prove the theorem in two steps. In the first step, we bound $F(\mathcal{S}\cup e) - F(\mathcal{S})$ using the proof of previous theorem, *i.e.*, Theorem 3. Hence, $F(\mathcal{S}\cup e) - F(\mathcal{S}) \geq \mathbb{E}[\log(\mu\cdot q)] + \log\left[\sqrt{2e\rho/\kappa_{\max,\lambda}^2}\right]$. Next we bound $F(\mathcal{T} \cup \{e\}) - F(\mathcal{T})$.

$$
\begin{aligned}
&F(\mathcal{T} \cup \{e\}) - F(\mathcal{T}) \\
&\overset{(i)}{\leq} \rho \left\|\boldsymbol{\theta}^*(\mathcal{H}_T\backslash(\mathcal{T}\cup\{e\}))\right\|^2 - \log\lambda_{\boldsymbol{\theta}^*(\mathcal{H}_T\backslash(\mathcal{T}\cup\{e\}))}(t\,|\,\mathcal{H}_t) \\
&\qquad - \log m_{\boldsymbol{\theta}^*(\mathcal{H}_T\backslash(\mathcal{T}\cup\{e\}))}(y\,|\,\mathcal{H}_t) + \mathbb{E}[\log(\mu\cdot q)] \\
&= \rho\left\|\boldsymbol{\theta}_\lambda^*(\mathcal{H}_T\backslash(\mathcal{T}\cup\{e\}))\right\|^2 - \log\left(\boldsymbol{\theta}_\lambda^*(\mathcal{H}_T\backslash(\mathcal{T}\cup\{e\}))^\top\boldsymbol{\kappa}_\lambda(t)\right) \\
&\qquad + \rho\left\|\boldsymbol{\theta}_m^*(\mathcal{H}_T\backslash(\mathcal{T}\cup\{e\}))\right\|^2 + \log\left[1 + \exp\left(-y\cdot\boldsymbol{\theta}_m^*(\mathcal{H}_T\backslash(\mathcal{T}\cup\{e\}))^\top\boldsymbol{\kappa}_m(t)\right)\right] \\
&\qquad + \mathbb{E}[\log(\mu\cdot q)] \tag{25}
\end{aligned}
$$

Here, inequality (i) is due to Lemma 10. Bounding $F(\mathcal{T} \cup \{e\}) - F(\mathcal{T})$ requires bounding the two terms in Eq. (25), *i.e.*, $\rho\left\|\boldsymbol{\theta}_\lambda^*(\mathcal{H}_T\backslash(\mathcal{T}\cup\{e\}))\right\|^2 - \log(\boldsymbol{\theta}_\lambda^*(\mathcal{H}_T\backslash(\mathcal{T}\cup\{e\}))^\top\boldsymbol{\kappa}_\lambda(t))$ and, $\rho\left\|\boldsymbol{\theta}_m^*(\mathcal{H}_T\backslash(\mathcal{T}\cup\{e\}))\right\|^2 + \log[1 + \exp\left(-y\cdot\boldsymbol{\theta}_m^*(\mathcal{H}_T\backslash(\mathcal{T}\cup\{e\}))^\top\boldsymbol{\kappa}_m(t)\right)]$.

> Bounding the first term *i.e.*, $\rho\left\|\boldsymbol{\theta}_\lambda^*(\mathcal{H}_T\backslash(\mathcal{T}\cup\{e\}))\right\|^2 - \log(\boldsymbol{\theta}_\lambda^*(\mathcal{H}_T\backslash(\mathcal{T}\cup\{e\}))^\top\boldsymbol{\kappa}_\lambda(t))$:

We note that

$$
\begin{aligned}
&\nabla_{\boldsymbol{\theta}_\lambda}\left[\rho\left|\mathcal{H}_T\backslash(\mathcal{T}\cup\{e\})\right|\cdot\left\|\boldsymbol{\theta}_\lambda\right\|^2\right. \\
&\qquad \left.\left. - \sum_{e_i\in\mathcal{H}_T\backslash(\mathcal{T}\cup\{e\})}\log(\boldsymbol{\theta}_\lambda^\top\boldsymbol{\kappa}_\lambda(t_i)) + \int_0^T\boldsymbol{\theta}_\lambda^\top\boldsymbol{\kappa}_\lambda(\tau)\,d\tau\right]\right|_{\boldsymbol{\theta}_\lambda=\boldsymbol{\theta}_\lambda^*(\mathcal{H}_T\backslash(\mathcal{T}\cup\{e\}))} = 0, \tag{26}
\end{aligned}
$$

as the corresponding maximum value occurs at $\boldsymbol{\theta}_\lambda = \boldsymbol{\theta}_\lambda^*(\mathcal{H}_T\backslash(\mathcal{T}\cup\{e\}))$. Hence, we have that:

$$
\begin{aligned}
&2\rho\left|\mathcal{H}_T\backslash(\mathcal{T}\cup\{e\})\right|\cdot\boldsymbol{\theta}_\lambda^*(\mathcal{H}_T\backslash(\mathcal{T}\cup\{e\})) \\
&\qquad - \sum_{e_i\in\mathcal{H}_T\backslash(\mathcal{T}\cup\{e\})}\frac{\boldsymbol{\kappa}_\lambda(t_i)}{\boldsymbol{\theta}_\lambda^*(\mathcal{H}_T\backslash(\mathcal{T}\cup\{e\}))^\top\boldsymbol{\kappa}_\lambda(t_i)} + \int_0^T\boldsymbol{\kappa}_\lambda(\tau)\,d\tau = 0 \tag{27}
\end{aligned}
$$

As, $\boldsymbol{\kappa}_\lambda(\cdot) \succeq 0$ ($\succeq$ indicates pointwise inequality), the above equation implies that:

$$
\begin{aligned}
2\rho\left|\mathcal{H}_T\backslash(\mathcal{T}\cup\{e\})\right|\cdot&\boldsymbol{\theta}_\lambda^*(\mathcal{H}_T\backslash(\mathcal{T}\cup\{e\})) \\
&\preceq \sum_{e_i\in\mathcal{H}_T\backslash(\mathcal{T}\cup\{e\})}\frac{\boldsymbol{\kappa}_\lambda(t_i)}{\boldsymbol{\theta}_\lambda^*(\mathcal{H}_T\backslash(\mathcal{T}\cup\{e\}))^\top\boldsymbol{\kappa}_\lambda(t_i)} \\
&\preceq \sum_{e_i\in\mathcal{H}_T\backslash(\mathcal{T}\cup\{e\})}\frac{\boldsymbol{\kappa}_\lambda(t_i)}{\left\|\boldsymbol{\theta}_\lambda^*(\mathcal{H}_T\backslash(\mathcal{T}\cup\{e\}))\right\|_1\boldsymbol{\kappa}_\lambda(t_i)} \\
&\preceq \sum_{e_i\in\mathcal{H}_T\backslash(\mathcal{T}\cup\{e\})}\frac{\boldsymbol{\kappa}_\lambda}{\left\|\boldsymbol{\theta}_\lambda^*(\mathcal{H}_T\backslash(\mathcal{T}\cup\{e\}))\right\|_1\,\min\limits_{v\in[d],e_i,e_j\in\mathcal{H}_T}\kappa_v(t_i,t_j)} \\
&\overset{(i)}{\preceq} \sum_{e_i\in\mathcal{H}_T\backslash(\mathcal{T}\cup\{e\})}\frac{\boldsymbol{\kappa}_\lambda}{\left\|\boldsymbol{\theta}_\lambda^*(\mathcal{H}_T\backslash(\mathcal{T}\cup\{e\}))\right\|_2\,\min\limits_{v\in[d],e_i,e_j\in\mathcal{H}_T}\kappa_v(t_i,t_j)} \tag{28}
\end{aligned}
$$

In each of the above, we consider pointwise inequality (or equality) by $\succeq$. (i) is due to the fact that $\|\boldsymbol{\theta}_\lambda\|_1 \geq \|\boldsymbol{\theta}_\lambda\|_2$. Eq. (28) further implies that,

$$2\rho\,|\mathcal{H}_T\backslash(\mathcal{T}\cup\{e\})|\cdot\|\boldsymbol{\theta}_\lambda^*(\mathcal{H}_T\backslash(\mathcal{T}\cup\{e\}))\|_2\,\boldsymbol{\theta}_\lambda^*(\mathcal{H}_T\backslash(\mathcal{T}\cup\{e\}))\min_{v\in[d],e_i,e_j\in\mathcal{H}_T}\kappa_v(t_i,t_j)\preceq\sum_{e_i\in\mathcal{H}_T\backslash(\mathcal{T}\cup\{e\})}\boldsymbol{\kappa}_\lambda$$

$$\implies 2\rho\,|\mathcal{H}_T\backslash(\mathcal{T}\cup\{e\})|\cdot\|\boldsymbol{\theta}_\lambda^*(\mathcal{H}_T\backslash(\mathcal{T}\cup\{e\}))\|_2^2\min_{v\in[d],e_i,e_j\in\mathcal{H}_T}\kappa_v(t_i,t_j)\leq\left\|\sum_{e_i\in\mathcal{H}_T\backslash(\mathcal{T}\cup\{e\})}\boldsymbol{\kappa}_\lambda\right\|$$

$$\leq |\mathcal{H}_T\backslash(\mathcal{T}\cup\{e\})|\,\kappa_{\max,\lambda}$$

$$\implies \|\boldsymbol{\theta}_\lambda^*(\mathcal{H}_T\backslash(\mathcal{T}\cup\{e\}))\|\leq\theta_{\max,\lambda}=\sqrt{\frac{\kappa_{\max,\lambda}}{2\rho\kappa_{\min,\lambda}}} \tag{29}$$

Eqs. (27) and (29) suggest that,

$$2\rho\,|\mathcal{H}_T\backslash(\mathcal{T}\cup\{e\})|\,\theta_{\max,\lambda}+K\geq\frac{\kappa_{\min,\lambda}\,|\mathcal{H}_T\backslash(\mathcal{T}\cup\{e\})|}{\|\boldsymbol{\theta}_\lambda^*(\mathcal{H}_T\backslash(\mathcal{T}\cup\{e\}))\|\,\kappa_{\max,\lambda}}$$

$$\implies\|\boldsymbol{\theta}_\lambda^*(\mathcal{H}_T\backslash(\mathcal{T}\cup\{e\}))\|\geq\theta_{\min,\lambda}=\frac{\kappa_{\min,\lambda}}{\kappa_{\max,\lambda}\left(2\rho\sqrt{\frac{\kappa_{\max,\lambda}}{2\rho\kappa_{\min,\lambda}}}+K\right)} \tag{30}$$

Hence, we bound $\rho\,\|\boldsymbol{\theta}_\lambda^*(\mathcal{H}_T\backslash(\mathcal{T}\cup\{e\}))\|^2-\log(\boldsymbol{\theta}_\lambda^*(\mathcal{H}_T\backslash(\mathcal{T}\cup\{e\}))^\top\boldsymbol{\kappa}_\lambda(t))$ as follows:

$$\rho\,\|\boldsymbol{\theta}_\lambda^*(\mathcal{H}_T\backslash(\mathcal{T}\cup\{e\}))\|^2-\log(\boldsymbol{\theta}_\lambda^*(\mathcal{H}_T\backslash(\mathcal{T}\cup\{e\}))^\top\boldsymbol{\kappa}_\lambda(t))$$

$$\leq\rho\,\theta_{\max,\lambda}^2-\log[\|\boldsymbol{\theta}_\lambda\|_1\,\kappa_{\min,\lambda}]$$

$$\overset{(i)}{\leq}\rho\,\theta_{\max,\lambda}^2-\log[\|\boldsymbol{\theta}_\lambda\|_2\,\kappa_{\min,\lambda}]$$

$$\leq\rho\,\theta_{\max,\lambda}^2-\log[\theta_{\min,\lambda}\,\kappa_{\min,\lambda}]$$

$$=\frac{\kappa_{\max,\lambda}}{2\kappa_{\min,\lambda}}+\log\left(\frac{\kappa_{\max,\lambda}}{\kappa_{\min,\lambda}^2}\right)+\log\left(K+\sqrt{\frac{2\rho\kappa_{\max,\lambda}}{\kappa_{\min,\lambda}}}\right). \tag{31}$$

Here inequality (i) is due to the fact that $\|\boldsymbol{\theta}_\lambda\|_1\geq\|\boldsymbol{\theta}_\lambda\|_2$.

---

Bounding the second term *i.e.*, $\rho\,\|\boldsymbol{\theta}_m^*(\mathcal{H}_T\backslash(\mathcal{T}\cup\{e\}))\|^2+\log[1+\exp\left(-y\cdot\boldsymbol{\theta}_m^*(\mathcal{H}_T\backslash(\mathcal{T}\cup\{e\}))^\top\boldsymbol{\kappa}_m(t)\right)]$:

First, we note that

$$\rho|\mathcal{H}_T\backslash(\mathcal{T}\cup\{e\})|\,\|\boldsymbol{\theta}_m^*(\mathcal{H}_T\backslash(\mathcal{T}\cup\{e\}))\|^2$$

$$\leq\rho|\mathcal{H}_T\backslash(\mathcal{T}\cup\{e\})|\,\|\boldsymbol{\theta}_m^*(\mathcal{H}_T\backslash(\mathcal{T}\cup\{e\}))\|^2$$

$$+\sum_{e_i\in\mathcal{H}_T\backslash(\mathcal{T}\cup\{e\})}\log[1+\exp\left(-y_i\cdot\boldsymbol{\theta}_m^*(\mathcal{H}_T\backslash(\mathcal{T}\cup\{e\}))^\top\boldsymbol{\kappa}_m(t_i)\right)]$$

$$\overset{(i)}{\leq}|\mathcal{H}_T\backslash(\mathcal{T}\cup\{e\})|\cdot\log 2$$

$$\implies\|\boldsymbol{\theta}_m^*(\mathcal{H}_T\backslash(\mathcal{T}\cup\{e\}))\|\leq\sqrt{\frac{\log 2}{\rho}}, \tag{32}$$

where (i) is due to the fact that:

$$\rho|\mathcal{H}_T\backslash(\mathcal{T}\cup\{e\})|\,\|\boldsymbol{\theta}_m^*(\mathcal{H}_T\backslash(\mathcal{T}\cup\{e\}))\|^2$$

$$+\sum_{e_i\in\mathcal{H}_T\backslash(\mathcal{T}\cup\{e\})}\log[1+\exp\left(-y_i\cdot\boldsymbol{\theta}_m^*(\mathcal{H}_T\backslash(\mathcal{T}\cup\{e\}))^\top\boldsymbol{\kappa}_m(t_i)\right)] \tag{33}$$

$$\leq\rho\times\mathbf{0}+\sum_{e_i\in\mathcal{H}_T\backslash(\mathcal{T}\cup\{e\})}\log[1+\exp\left(-y_i\cdot\mathbf{0}^\top\boldsymbol{\kappa}_m(t_i)\right)]=|\mathcal{H}_T\backslash(\mathcal{T}\cup\{e\})|\cdot\log 2 \tag{34}$$

which is because, $\boldsymbol{\theta}_m^*(\mathcal{H}_T\backslash(\mathcal{T}\cup\{e\}))$ is the minimum of Eq. (33).

Hence, we have that:

$$\rho \left\|\boldsymbol{\theta}_m^*(\mathcal{H}_T\backslash(\mathcal{T}\cup\{e\}))\right\|^2 + \log\left[1 + \exp\left(-y\cdot\boldsymbol{\theta}_m^*(\mathcal{H}_T\backslash(\mathcal{T}\cup\{e\}))^\top\boldsymbol{\kappa}_m(t)\right)\right]$$

$$\leq \rho\left\|\boldsymbol{\theta}_m^*(\mathcal{H}_T\backslash(\mathcal{T}\cup\{e\}))\right\|^2 + \log 2 + \boldsymbol{\theta}_m^*(\mathcal{H}_T\backslash(\mathcal{T}\cup\{e\}))^\top\boldsymbol{\kappa}_m(t) + (\boldsymbol{\theta}_m^*(\mathcal{H}_T\backslash(\mathcal{T}\cup\{e\}))^\top\boldsymbol{\kappa}_m(t))^2/2$$

$$\leq \rho\left\|\boldsymbol{\theta}_m^*(\mathcal{H}_T\backslash(\mathcal{T}\cup\{e\}))\right\|^2 + \log 2 + \left\|\boldsymbol{\theta}_m^*(\mathcal{H}_T\backslash(\mathcal{T}\cup\{e\}))\right\|\kappa_{\max,m} + \left\|\boldsymbol{\theta}_m^*(\mathcal{H}_T\backslash(\mathcal{T}\cup\{e\}))\right\|^2\kappa_{\max,m}^2/2$$

$$\leq 2\log 2 + \kappa_{\max,m}^2\log 2/(2\rho) + \kappa_{\max,m}\sqrt{\log 2/\rho} \tag{35}$$

First we add the bounds in Eqs. (31) and (35) and then put it into Eq. (25) to obtain the required bound on $F(\mathcal{H}_T\backslash(\mathcal{T}\cup\{e\})) - F(\mathcal{T})$. The above result together with the bound on $F(\mathcal{S}\cup e) - F(\mathcal{S})$ gives us the required bound on $\alpha$.

## C.2   Monotonicity and approximation guarantee for nonlinear MTPP

**Theorem 5** *Assume that the endogenous events follow a nonlinear MTPP with exponentially bounded intensity function, as described in Eq. (10) with $\lambda_{\max} \leq \exp(-\mathbb{E}[\log(\mu\cdot q)])$. Then the set function $F(\mathcal{S})$ defined in Eq. (7) is monotone in $\mathcal{S}$ if $\rho \geq 0.25\,a^2/\left(\log\left[1/\lambda_{\max}\right] + \mathbb{E}[\log(\mu\cdot q)]\right)$, where $\mu$ and $q$ are the intensity and the mark distribution for the exogenous events.*

**Proof.** From Lemma 9 in Appendix D, we have that: for $e = (y,t) \notin \mathcal{S}$,

$$F(\mathcal{S}\cup\{e\}) - F(\mathcal{S}) \geq \rho\left\|\boldsymbol{\theta}^*(\mathcal{H}_T\backslash\mathcal{S})\right\|^2 - \log\lambda_{\boldsymbol{\theta}^*(\mathcal{H}_T\backslash\mathcal{S})}(t\,|\,\mathcal{H}_t) + \mathbb{E}[\log(\mu\cdot q)]$$

$$\geq \min_{\boldsymbol{\theta}}[\rho\left\|\boldsymbol{\theta}\right\|^2 - \log\lambda_{\max} - a\left\|\boldsymbol{\theta}\right\|] + \mathbb{E}[\log(\mu\cdot q)]$$

$$\geq -\frac{a^2}{4\rho} + \left[\log\left(\frac{1}{\lambda_{\max}}\right) + \mathbb{E}[\log\mu + \log q]\right]$$

$$> 0 \tag{36}$$

The last inequality follows from the assumption of $\rho$. This readily proves the theorem.

**Theorem 6** *Given the same conditions in Theorem 5 and $\int_0^\infty \left\|\lambda_\mathbf{0}(t\,|\,\mathcal{H}_t)\right\|_2 dt \leq \Lambda_{\max} < \infty$, the set function $F(\mathcal{S})$ is $\alpha$-submodular in $\mathcal{S}$ with*

$$\alpha \geq \alpha_F = \frac{\log\left[\exp\left(-a^2\mathbb{E}[\log(\mu\cdot q)]/4\right)/\lambda_{\max}\right]}{\Lambda_{\max} + (a+b)\sqrt{\dfrac{\log(1/\lambda_{\min}m_{\min}) + \Lambda_{\max}}{\rho}} + 2\log(1/\lambda_{\min}m_{\min}) + \mathbb{E}[\log(\mu\cdot q)]} \tag{37}$$

**Proof.** In the first step, we bound $F(\mathcal{S}\cup e) - F(\mathcal{S})$ using the proof of previous theorem, *i.e.*, Theorem 5. Hence, $F(\mathcal{S}\cup e) - F(\mathcal{S}) \geq \log\left[\exp\left(-a^2\mathbb{E}[\log(\mu\cdot q)]/4\right)/\lambda_0\right]$. Next we bound $F(\mathcal{T}\cup\{e\}) - F(\mathcal{T})$. We observe that:

$$F(\mathcal{T}\cup\{e\}) - F(\mathcal{T})$$

$$\overset{(i)}{\leq} \rho\left\|\boldsymbol{\theta}^*(\mathcal{H}_T\backslash(\mathcal{T}\cup\{e\}))\right\|^2 - \log\lambda_{\boldsymbol{\theta}^*(\mathcal{H}_T\backslash(\mathcal{T}\cup\{e\}))}(t\,|\,\mathcal{H}_t)$$

$$\quad - \log m_{\boldsymbol{\theta}^*(\mathcal{H}_T\backslash(\mathcal{T}\cup\{e\}))}(y\,|\,\mathcal{H}_t) + \mathbb{E}[\log(\mu\cdot q)]$$

$$\overset{(ii)}{\leq} \rho\left\|\boldsymbol{\theta}^*(\mathcal{H}_T\backslash(\mathcal{T}\cup\{e\}))\right\|^2 + (a+b)\left\|\boldsymbol{\theta}^*(\mathcal{H}_T\backslash(\mathcal{T}\cup\{e\}))\right\| - \log(\lambda_{\min}m_{\min}) + \mathbb{E}[\log(\mu\cdot q)] \tag{38}$$

Here, inequality (i) is due to Lemma 10 and inequality (ii) comes from the fact that $\lambda \geq \lambda_0\exp(-a\left\|\boldsymbol{\theta}\right\|)$ and $m \geq \exp(-b\left\|\boldsymbol{\theta}\right\|)$. In order to bound Eq. (38), we bound $\left\|\boldsymbol{\theta}\right\|$. To this aim, we have:

$$\rho\left\|\theta^*(\mathcal{H}_T\backslash(\mathcal{T}\cup\{e\}))\right\|^2|\mathcal{H}_T\backslash(\mathcal{T}\cup\{e\})|$$

$$\leq \rho\left\|\theta^*(\mathcal{H}_T\backslash(\mathcal{T}\cup\{e\}))\right\|^2|\mathcal{H}_T\backslash(\mathcal{T}\cup\{e\})|$$

$$\quad - \sum_{e_i\in\mathcal{H}_T\backslash(\mathcal{T}\cup\{e\})}\log\left[\lambda_{\theta^*(\mathcal{H}_T\backslash(\mathcal{T}\cup\{e\}))}(t_i\,|\,\mathcal{H}_{t_i})\cdot m_{\theta^*(\mathcal{H}_T\backslash(\mathcal{T}\cup\{e\}))}(y_i\,|\,\mathcal{H}_{t_i})\right]$$

$$\quad + \int_0^T \lambda_{\theta^*(\mathcal{H}_T\backslash(\mathcal{T}\cup\{e\}))}(\tau\,|\,\mathcal{H}_\tau)\,d\tau \tag{39}$$

which is less than the value of the objective given by $\boldsymbol{\theta} = \boldsymbol{0}$. Thus,

$$\rho \left\| \theta^*(\mathcal{H}_T \backslash (\mathcal{T} \cup \{e\})) \right\|^2 |\mathcal{H}_T \backslash (\mathcal{T} \cup \{e\})| \leq |\mathcal{H}_T \backslash (\mathcal{T} \cup \{e\})| \log(1/\lambda_{\min} m_{\min}) + \Lambda_{\max}$$

$$\implies \left\| \theta^*(\mathcal{H}_T \backslash (\mathcal{T} \cup \{e\})) \right\|^2 \leq \frac{\log(1/\lambda_{\min} m_{\min}) + \Lambda_{\max}}{\rho} \tag{40}$$

To this end, we have:

$$F(\mathcal{T} \cup \{e\}) - F(\mathcal{T}) \leq \Lambda_{\max} + (a+b)\sqrt{\frac{\log(1/\lambda_{\min} m_{\min}) + \Lambda_{\max}}{\rho}}$$

$$+ 2\log(1/\lambda_{\min} m_{\min}) + \mathbb{E}[\log(\mu \cdot q)] \tag{41}$$

The above result together with the bound on $F(\mathcal{S} \cup e) - F(\mathcal{S})$ gives us the required bound.

### C.3 Representation of MTPPs using exponential bounds

Here, we discuss that under some mild conditions, popular MTPP models [17, 89] can admit exponential bounds when the hidden states $\boldsymbol{h}_i$ are bounded, *i.e.*, $||\boldsymbol{h}_i|| < h_{\max}$. Furthermore, we assume that $t_i - t_{i-1} \leq \delta_{\max}$, $t_0 > t_{\min} > 0$.

**Exponential boundedness of RMTPP.** RMTPP [17] considers exponential intensity function, *i.e.*,

$$\lambda_{\boldsymbol{\theta}}(t_{i+1} \mid \mathcal{H}_{t_i}) = \exp(\boldsymbol{w}^\top \boldsymbol{h}_i + \nu(t - t_i) + v). \tag{42}$$

Then we have:

$$\exp(-(h_{\max} + \delta_{\max} + 1)\|[\boldsymbol{w}, \nu, v]\|_2) \leq \lambda_{\boldsymbol{\theta}}(t_{i+1} \mid \mathcal{H}_{t_i}) \leq \exp((h_{\max} + \delta_{\max} + 1)\|[\boldsymbol{w}, \nu, v]\|_2) \tag{43}$$

**Exponential boundedness of THP.** In Transformer Hawkes Process [89], a softplus activation function is used for intensity modeling. More specifically, we have:

$$\lambda_{\boldsymbol{\theta}}(t_{i+1} \mid \mathcal{H}_{t_i}) = \sum_{j=1}^{J} \text{Softplus}\left(\alpha_j \frac{t - t_i}{t_i} + \boldsymbol{w}_j^\top \boldsymbol{h}_i + v_j\right) \tag{44}$$

In the following, we formally prove the exponential bounds for THP.

**Proposition 8** *Given a Transformer Hawkes Process [89] with the intensity function* (44). *If in Eq.* (44)*, we have* $u_j = \alpha_j \frac{t - t_i}{t_i} + \boldsymbol{w}_j^\top \boldsymbol{h}_i + v_j \geq u_{\min} > 0$*; the hidden states* $\boldsymbol{h}_i$ *are bounded,* i.e., $||\boldsymbol{h}_i|| < h_{\max}$*; and,* $t_i - t_{i-1} \leq \delta_{\max}$*,* $t_0 > t_{\min} > 0$*, then we have:*

$$\lambda_0 \exp(-a\|\boldsymbol{\theta}\|_2) \leq \lambda_{\boldsymbol{\theta}}(t_{i+1} \mid \mathcal{H}_{t_i}) \leq \lambda_0 \exp(a\|\boldsymbol{\theta}\|_2) \tag{45}$$

*where,* $\lambda_0 = J$*,* $a = \sqrt{D}(2 + \log 2/u_{\min})(\delta_{\max}/t_{\min} + h_{\max} + 1)$*.*

**Proof.** Define $\omega = 2 + \log 2/u_{\min}$ Due to the fact that $1 - 1/x \leq \log(x) \leq x - 1$, we have:

$$\frac{\exp(u_j)}{1 + \exp(u_j)} \leq \text{Softplus}(u_j) \leq \exp(u_j)$$

$$\overset{(i)}{\implies} 0.5 \cdot \exp(-2|u_j|) \leq \text{Softplus}(u_j) \leq \exp(2|u_j|),$$

$$\overset{(ii)}{\implies} \exp(-\omega|u_j|) \leq \text{Softplus}(u_j) \leq \exp(\omega|u_j|),$$

$$\implies \sum_{j=1}^{J} \exp(-\omega|u_j|) \leq \lambda_{\boldsymbol{\theta}}(t_{i+1} \mid \mathcal{H}_{t_i}) \leq \sum_{j=1}^{J} \exp(\omega|u_j|), \tag{46}$$

where (i) is due to the fact that

$$\exp(-|u_j|)(\exp(u_j) + 1) \leq 2 \implies \frac{\exp(-|u_j|)}{2} \leq \frac{1}{1 + \exp(u_j)} \text{ and } \exp(-|u_j|) \leq \exp(u_j),$$

(ii) is due to the fact that

$$\exp(-2|u_j| - (|u_j|/u_{\min})\log 2) \leq \frac{1}{2}\exp(-2|u_j|) \tag{47}$$

Now we have the following bounds.

$$\sum_{j=1}^{J} \exp(\omega|u_j|) \leq \sum_{j=1}^{J} \exp\left(\omega\left|\alpha_j\frac{t-t_i}{t_i} + \boldsymbol{w}_j^\top \boldsymbol{h}_i + v_j\right|\right)$$

$$\overset{(i)}{\leq} J\exp\left(\omega\sum_{j=1}^{J}\|[\boldsymbol{w}_j,\ \alpha_j,\ v_j]\|_1\left(\delta_{\max}/t_{\min} + h_{\max} + 1\right)\right)$$

$$\overset{(ii)}{\leq} J\exp\left(\omega\left\|\odot_{j=1}^{J}[\boldsymbol{w}_j,\ \alpha_j,\ v_j]\right\|_1\left(\delta_{\max}/t_{\min} + h_{\max} + 1\right)\right)$$

$$\overset{(iii)}{\leq} J\exp\left(\sqrt{D}\omega\left\|\odot_{j=1}^{J}[\boldsymbol{w}_j,\ \alpha_j,\ v_j]\right\|_2\left(\delta_{\max}/t_{\min} + h_{\max} + 1\right)\right). \qquad (48)$$

Here $D$ is the total number of parameters, *i.e.*, $D = J(\dim(\boldsymbol{w}) + 2)$; $\odot$ indicates concatenation; inequality (i) is due to that $|\boldsymbol{a}\cdot\boldsymbol{b}| < \|\boldsymbol{a}\|_1 b_{\max}$, inequality (ii) is due to that $\sum_i\|\boldsymbol{a}_i\|_1 = \|\odot_i\boldsymbol{a}_i\|_1$ and inequality (iii) is due to the fact that $\|\boldsymbol{a}\|_1 \leq \sqrt{\dim(\boldsymbol{a})}\|\boldsymbol{a}\|_2$. Moreover, we have:

$$\sum_{j=1}^{J}\exp(-\omega|u_j|) \overset{(i)}{\geq} J\exp\left(-\omega\sum_{j=1}^{J}\left|\alpha_j\frac{t-t_i}{t_i} + \boldsymbol{w}_j^\top\boldsymbol{h}_i + v_j\right|/J\right)$$

$$\overset{(ii)}{\geq} J\exp\left(-\omega\sum_{j=1}^{J}\|[\boldsymbol{w}_j,\ \alpha_j,\ v_j]\|_1\left(\delta_{\max}/t_{\min} + h_{\max} + 1\right)/J\right)$$

$$\overset{(iii)}{\geq} J\exp\left(-\omega\left\|\odot_{j=1}^{J}[\boldsymbol{w}_j,\ \alpha_j,\ v_j]\right\|_1\left(\delta_{\max}/t_{\min} + h_{\max} + 1\right)/J\right)$$

$$\overset{(iv)}{\geq} J\exp\left(-\sqrt{D}\omega\left\|\odot_{j=1}^{J}[\boldsymbol{w}_j,\ \alpha_j,\ v_j]\right\|_2\left(\delta_{\max}/t_{\min} + h_{\max} + 1\right)/J\right),$$

$$\geq J\exp\left(-\sqrt{D}\omega\left\|\odot_{j=1}^{J}[\boldsymbol{w}_j,\ \alpha_j,\ v_j]\right\|_2\left(\delta_{\max}/t_{\min} + h_{\max} + 1\right)\right), \qquad (49)$$

where (i) is due to A.M $\geq$ G.M, inequality (ii) is due to that $|\boldsymbol{a}\cdot\boldsymbol{b}| < \|\boldsymbol{a}\|_1 b_{\max}$, inequality (iii) is due to that $\sum_i\|\boldsymbol{a}_i\|_1 = \|\odot_i\boldsymbol{a}_i\|_1$ and inequality (iv) is due to the fact that $\|\boldsymbol{a}\|_1 \leq \sqrt{\dim(\boldsymbol{a})}\|\boldsymbol{a}\|_2$. Eqs. (48) and (49) give us required parameters for the exponential bounds, *i.e.*, $\lambda_0$ and $a$ in Eq. (45).

### C.4 Approximation guarantees of Algorithm 1

We first restate the Theorem 7.

**Theorem 7** *Assume that the training algorithm* TRAIN() *in Algorithm 1 provides imperfect estimates of the underlying model parameters, with $\mathcal{L}(\boldsymbol{\theta}^*(\mathcal{H}_T\backslash\mathcal{S}); \mathcal{S}; \mathcal{H}_T) - \mathcal{L}(\widehat{\boldsymbol{\theta}}(\mathcal{H}_T\backslash\mathcal{S}); \mathcal{S}; \mathcal{H}_T)) \leq \epsilon$ for all $\mathcal{S}$ and that it runs with $|\mathcal{V}| = O\left(\frac{|\mathcal{H}_T|}{k}\log(1/\delta)\right)$ (cf. line number 3). Then,*

$$\mathbb{E}[F(\mathcal{S})] \geq (1 - \exp(-\alpha_F^*) - \alpha_F^*\cdot\delta)\cdot F(\mathcal{S}^*) - k\epsilon \qquad (50)$$

*where the expectation is taken over many draws of $\mathcal{V}$ (cf. line number 3), $\mathcal{S}^*$ is the solution of the optimization problem* (8) *and $\alpha_F^*$ is the submodularity ratio, computed using Theorems 4 and 5.*

**Proof** Through the rest of this proof, we denote $F(\mathcal{S}) = \mathcal{L}(\boldsymbol{\theta}^*(\mathcal{H}_T\backslash\mathcal{S}); \mathcal{S}; \mathcal{H}_T)$ and $\hat{F}(\mathcal{S}) = \mathcal{L}(\widehat{\boldsymbol{\theta}}(\mathcal{H}_T\backslash\mathcal{S}); \mathcal{S}; \mathcal{H}_T))$. So essentially, we assume we do not have access to $F$ but rather an approximation $\hat{F}$. In other words, we pick $i \in \operatorname{argmax}_{j\in\mathcal{V}\backslash\mathcal{S}_i}\hat{F}(j|\mathcal{S}_i)$. Note that for any set $\mathcal{S}$, $|F(\mathcal{S}) - \hat{F}(\mathcal{S})| \leq \epsilon$. Hence, it implies that $F(\mathcal{S}) - \epsilon \leq \hat{F}(\mathcal{S}) \leq F(\mathcal{S}) + \epsilon$. Given this, we have that $F(i|\mathcal{S}) - 2\epsilon \leq \hat{F}(i|\mathcal{S}) = \hat{F}(\mathcal{S}\cup i) - \hat{F}(\mathcal{S}) \leq F(i|\mathcal{S}) + 2\epsilon$. Now given a $\hat{i}$ such that $\hat{F}(\hat{i}|\mathcal{S}) \geq \hat{F}(i|\mathcal{S}), \forall i \notin \mathcal{S}$, we have that $F(\hat{i}|\mathcal{S}) - 2\epsilon \geq F(i|\mathcal{S}) + 2\epsilon$ which implies that:

$$F(\hat{i}|\mathcal{S}) \geq F(i|\mathcal{S}) + 4\epsilon \qquad (51)$$

Next, we study the bound for the stochastic greedy algorithm (Algorithm 1). Following Lemma 2 from [55], at every round of the stochastic greedy algorithm, and combining with Equ. (51), we have:

$$\mathbb{E}(F(a_{i+1}|\mathcal{S}_i)) \geq \frac{1-\delta}{k} \sum_{a \in \mathcal{S}^* \setminus \mathcal{S}_i} [F(a|\mathcal{S}_i) - 4\epsilon] \tag{52}$$

Here $a_{i+1}$ is the element picked at the $i$th round of stochastic greedy. Also, $\mathcal{S}_i$ is the set obtained by stochastic greedy in the $i$th step of greedy. Following from $\alpha$-submodularity, we have that:

$$\sum_{a \in \mathcal{S}^* \setminus \mathcal{S}_i} F(a|\mathcal{S}_i) \geq \alpha F(\mathcal{S}^*|\mathcal{S}_i) \geq \alpha(F(\mathcal{S}^*) - F(\mathcal{S}_i)) \tag{53}$$

This is because while picking the best item in the stochastic greedy algorithm, we have access only to $\hat{F}$ and not $F$. This implies,

$$\mathbb{E}[F(\mathcal{S}_{i+1}) - F(\mathcal{S}_i)] \geq [(1-\delta)\alpha - 4k\epsilon/\text{OPT}]/k\mathbb{E}[F(\mathcal{S}^*) - F(\mathcal{S}_i)] \tag{54}$$

Here, we denote $F(\mathcal{S}^*) = \text{OPT}$. Next, define $\Gamma_i = \mathbb{E}[f(\mathcal{S}^*) - f(\mathcal{S}_i)]$. We then have the following recursion:

$$\Gamma_i - \Gamma_{i+1} \geq [(1-\delta)\alpha - 4k\epsilon/\text{OPT}]/k\Gamma_i \tag{55}$$

which implies that $\Gamma_{i+1} \leq \{1 - [(1-\delta)\alpha - 4k\epsilon/\text{OPT}]/k\}$. Using induction, we get:

$$\mathbb{E}[F(\mathcal{S}_k)] \geq (1 - (1 - \frac{(1-\delta)\alpha}{k} - 4\epsilon/\text{OPT})^k)F(\mathcal{S}^*) \tag{56}$$

Since $\delta, \epsilon \approx 0$, this gives us an approximation factor of $1 - e^{-\alpha} - \alpha\delta - k\epsilon/OPT$.

Combining everything, this proves the result:

$$\mathbb{E}[F(\mathcal{S})] \geq (1 - \exp(-\alpha_F^*) - \alpha_F^* \cdot \delta) \cdot F(\mathcal{S}^*) - k\epsilon \tag{57}$$

∎

# D   Additional theoretical results

**Lemma 9** *Given that $F(\mathcal{S})$ is defined in Eq. (7), then for any $e = (y, t) \in \mathcal{H}_T \backslash \mathcal{S}$, we have:*

$$F(\mathcal{S} \cup \{e\}) - F(\mathcal{S}) \geq \rho \left\| \boldsymbol{\theta}^*(\mathcal{H}_T \backslash \mathcal{S}) \right\|^2 - \log \lambda_{\boldsymbol{\theta}^*(\mathcal{H}_T \backslash \mathcal{S})}(t \,|\, \mathcal{H}_t) + \mathbb{E}[\log(\mu \cdot q)]. \qquad (58)$$

*Here $\lambda_{\boldsymbol{\theta}}$ and $m_{\boldsymbol{\theta}}$ can take any functional form.*

**Proof.** Let us first define:

$$\mathcal{L}_\lambda \left( \boldsymbol{\theta}, \mathcal{S}; \mathcal{H}_T \right) = -\rho |\mathcal{H}_T \backslash \mathcal{S}| \, \|\boldsymbol{\theta}\|^2 + \sum_{e_i \in \mathcal{H}_T \backslash \mathcal{S}} \log \lambda_{\boldsymbol{\theta}}(t_i \,|\, \mathcal{H}_{t_i}) - \int_0^T \lambda_{\boldsymbol{\theta}}(\tau \,|\, \mathcal{H}_\tau) \, d\tau \qquad (59)$$

$$\mathcal{L}_p \left( \boldsymbol{\theta}, \mathcal{S}; \mathcal{H}_T \right) = \sum_{e_i \in \mathcal{H}_T \backslash \mathcal{S}} \log m_{\boldsymbol{\theta}}(y_i \,|\, \mathcal{H}_{t_i}), \qquad (60)$$

Hence, we have that $\mathcal{L}(\bullet) = \mathcal{L}_\lambda(\bullet) + \mathcal{L}_p(\bullet)$

$$F(\mathcal{S} \cup \{e\}) - F(\mathcal{S})$$

$$= \mathcal{L} \left( \boldsymbol{\theta}^*(\mathcal{H}_T \backslash (\mathcal{S} \cup \{e\})), \mathcal{S} \cup \{e\}; \mathcal{H}_T \right) - \mathcal{L} \left( \boldsymbol{\theta}^*(\mathcal{H}_T \backslash \mathcal{S}), \mathcal{S}; \mathcal{H}_T \right) + \mathbb{E}[\log(\mu \cdot q)]$$

$$= \underbrace{\mathcal{L} \left( \boldsymbol{\theta}^*(\mathcal{H}_T \backslash (\mathcal{S} \cup \{e\})), \mathcal{S} \cup \{e\}; \mathcal{H}_T \right) - \mathcal{L} \left( \boldsymbol{\theta}^*(\mathcal{H}_T \backslash \mathcal{S}), \mathcal{S} \cup \{e\}; \mathcal{H}_T \right)}_{\geq 0}$$

$$+ \mathcal{L} \left( \boldsymbol{\theta}^*(\mathcal{H}_T \backslash \mathcal{S}), \mathcal{S} \cup \{e\}; \mathcal{H}_T \right) - \mathcal{L} \left( \boldsymbol{\theta}^*(\mathcal{H}_T \backslash \mathcal{S}), \mathcal{S}; \mathcal{H}_T \right) + \mathbb{E}[\log(\mu \cdot q)]$$

$$\overset{(i)}{\geq} \Bigg[ \mathcal{L}_\lambda \left( \boldsymbol{\theta}^*(\mathcal{H}_T \backslash \mathcal{S}), \mathcal{S} \cup \{e\}; \mathcal{H}_T \right) - \mathcal{L}_\lambda \left( \boldsymbol{\theta}^*(\mathcal{H}_T \backslash \mathcal{S}), \mathcal{S}; \mathcal{H}_T \right)$$

$$+ \mathcal{L}_p \left( \boldsymbol{\theta}^*(\mathcal{H}_T \backslash \mathcal{S}), \mathcal{S} \cup \{e\}; \mathcal{H}_T \right) - \mathcal{L}_p \left( \boldsymbol{\theta}^*(\mathcal{H}_T \backslash \mathcal{S}), \mathcal{S}; \mathcal{H}_T \right) \Bigg] + \mathbb{E}[\log(\mu \cdot q)]$$

$$= \rho \left\| \boldsymbol{\theta}^*(\mathcal{H}_T \backslash \mathcal{S}) \right\|^2 - \log \lambda_{\boldsymbol{\theta}^*(\mathcal{H}_T \backslash \mathcal{S})}(t \,|\, \mathcal{H}_t) - \log m_{\boldsymbol{\theta}^*(\mathcal{H}_T \backslash \mathcal{S})})(y \,|\, \mathcal{H}_t) + \mathbb{E}[\log(\mu \cdot q)]$$

$$\overset{(ii)}{\geq} \rho \left\| \boldsymbol{\theta}^*(\mathcal{H}_T \backslash \mathcal{S}) \right\|^2 - \log \lambda_{\boldsymbol{\theta}^*(\mathcal{H}_T \backslash \mathcal{S})}(t \,|\, \mathcal{H}_t) + \mathbb{E}[\log(\mu \cdot q)], \qquad (61)$$

where (i) is due to the fact that $\boldsymbol{\theta}^*(\mathcal{H}_T \backslash (\mathcal{S} \cup \{e\})) = \operatorname{argmax}_{\boldsymbol{\theta}} \mathcal{L} \left( \boldsymbol{\theta}, \mathcal{S} \cup \{e\}; \mathcal{H}_T \right)$; (ii) is due to the fact that $m_{\boldsymbol{\theta}}$ is a probability distribution.

**Lemma 10** *Given that $F(\cdot)$ is defined in Eq. (7), then for any $e = (y, t) \in \mathcal{H}_T \backslash \mathcal{T}$, we have:*

$$F(\mathcal{T} \cup \{e\}) - F(\mathcal{T}) \leq \rho \left\| \boldsymbol{\theta}^*(\mathcal{H}_T \backslash (\mathcal{T} \cup \{e\})) \right\|^2 - \log \lambda_{\boldsymbol{\theta}^*(\mathcal{H}_T \backslash (\mathcal{T} \cup \{e\}))}(t \,|\, \mathcal{H}_t)$$

$$- \log m_{\boldsymbol{\theta}^*(\mathcal{H}_T \backslash (\mathcal{T} \cup \{e\}))}(y \,|\, \mathcal{H}_t) + \mathbb{E}[\log(\mu \cdot q)] \qquad (62)$$

*Here $\lambda_{\boldsymbol{\theta}}$ and $m_{\boldsymbol{\theta}}$ can take any functional form.*

**Proof.** We note that:

$$F(\mathcal{T} \cup \{e\}) - F(\mathcal{T})$$

$$= \mathcal{L} \left( \boldsymbol{\theta}^*(\mathcal{H}_T \backslash (\mathcal{T} \cup \{e\})), \mathcal{T} \cup \{e\}; \mathcal{H}_T \right) - \mathcal{L} \left( \boldsymbol{\theta}^*(\mathcal{H}_T \backslash \mathcal{T}), \mathcal{T}; \mathcal{H}_T \right) + \mathbb{E}[\log(\mu \cdot q)]$$

$$= \mathcal{L} \left( \boldsymbol{\theta}^*(\mathcal{H}_T \backslash (\mathcal{T} \cup \{e\})), \mathcal{T} \cup \{e\}; \mathcal{H}_T \right) - \mathcal{L} \left( \boldsymbol{\theta}^*(\mathcal{H}_T \backslash (\mathcal{T} \cup \{e\})), \mathcal{T}; \mathcal{H}_T \right)$$

$$+ \underbrace{\mathcal{L} \left( \boldsymbol{\theta}^*(\mathcal{H}_T \backslash (\mathcal{T} \cup \{e\})), \mathcal{T}; \mathcal{H}_T \right) - \mathcal{L} \left( \boldsymbol{\theta}^*(\mathcal{H}_T \backslash \mathcal{T}), \mathcal{T}; \mathcal{H}_T \right)}_{\leq 0} + \mathbb{E}[\log(\mu \cdot q)]$$

$$\overset{(i)}{\leq} \rho \left\| \boldsymbol{\theta}^*(\mathcal{H}_T \backslash (\mathcal{T} \cup \{e\})) \right\|^2$$

$$- \log \lambda_{\boldsymbol{\theta}^*(\mathcal{H}_T \backslash (\mathcal{T} \cup \{e\}))}(t \,|\, \mathcal{H}_t) - \log m_{\boldsymbol{\theta}^*(\mathcal{H}_T \backslash (\mathcal{T} \cup \{e\}))}(y \,|\, \mathcal{H}_t) + \mathbb{E}[\log(\mu \cdot q)]. \qquad (63)$$

Here, inequality (i) is due to

$$\boldsymbol{\theta}^*(\mathcal{H}_T \backslash \mathcal{T}) = \operatorname*{argmax}_{\boldsymbol{\theta}} \mathcal{L} \left( \boldsymbol{\theta}, \mathcal{T}; \mathcal{H}_T \right).$$

**Proposition 11** $\min_x (ax^2 - \log x) = \frac{1}{2} + \log(\sqrt{2a})$.

**Proof.** Differentiating with respect to $x$, we have: $x = 1/\sqrt{2a}$, which readily gives the result.

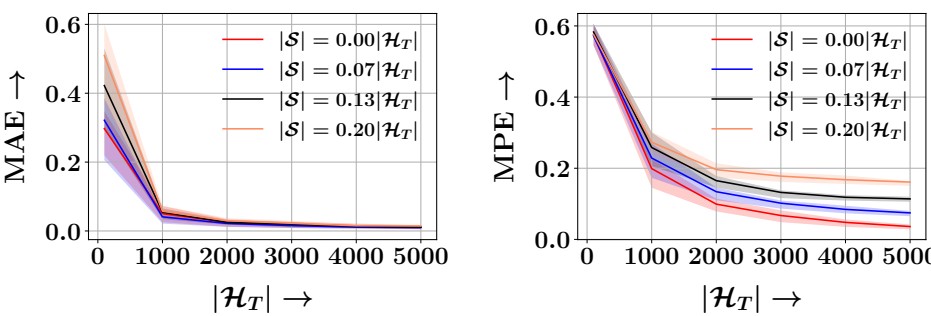

Figure 3: Variation of mean absolute error $\mathbb{E}(|t - \hat{t}|)$ and mark prediction error (MPE) $\Pr(y \neq \hat{y})$ against the number of observations $|\mathcal{H}_T|$ on synthetic data.

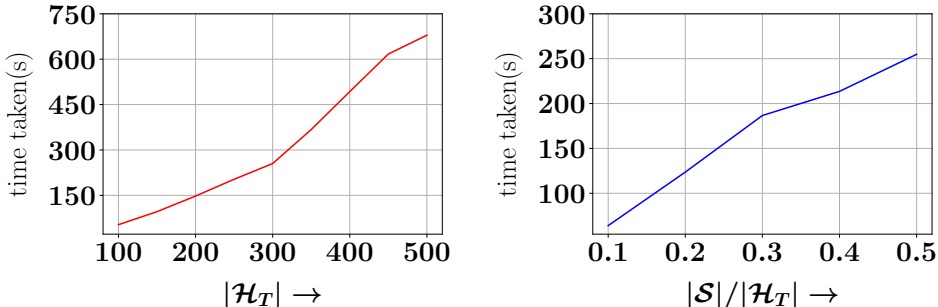

Figure 4: Scalability analysis. Variation of time taken against $|\mathcal{H}_T|$ and $|\mathcal{S}|/|\mathcal{H}_T|$. In the left figure, we set $|\mathcal{S}|/|\mathcal{H}_T| = 0.3$ and in the right figure, we set $|\mathcal{H}_T| = 200$.

## E  Additional experiments on synthetic data

### E.1  Quantitative analysis

Here, we investigate the variation of performance against the number of observations $|\mathcal{H}_T|$ on synthetic data for various values of $k$, the prespecified number of exogenous events. Figure 3 summarizes the results, which shows that, (i) as the number of observed events are increasing, the prediction error decreases; and, (ii) as $k$ increases, the prediction error increases.

### E.2  Scalability analysis

Next, we analyze the time taken by our method against $|\mathcal{H}_T|$— the total number of observations and $|\mathcal{S}|/|\mathcal{H}_T|$, the prespecified fraction of exogenous events. Figure 4 summarizes the figure which shows that, for a sequence with $|\mathcal{H}_T| = 500$, our method takes less than 13 minute to converge in single AMD Ryzen 7 3700X CPU with 32GB RAM machine.

| Dataset | $|\mathcal{H}_T|$ | $|\mathcal{C}|$ | Class imbalance factor |
|---|---|---|---|
| Club | 9409 | 3 | 3.88 |
| Election | 1584 | 3 | 2.62 |
| Series | 16318 | 3 | 1.83 |
| Verdict | 10691 | 3 | 2.13 |
| BookOrder | 15000 | 2 | 1.04 |

Table 4: Datasets statistics. Class imbalance factor indicates the ratio of highest to lowest occurring marks.

## F   Additional details about the experiments with real data

### F.1   Dataset details

We use five datasets in our experiments for evaluation. The first four datasets were used in previous work [84, 41, 17].

1. **Club** [84, 41]: It gathers the tweets on Barcelona getting the first place in La-liga, from May 8 to May 16, 2016.

2. **Election** [84, 41]: It gathers the tweets on British election, from May 7 to May 15, 2015.

3. **Series** [84]: It gathers the tweets on promotion on the TV show "Games of Thrones", from May 4 to May 12, 2015.

4. **Verdict** [41]: It gathers the tweets on the verdict for the corruption-case against Jayalalitha, an Indian politician, from May 6 to May 17, 2015.

5. **BookOrder** [17]: It gathers the limited book order data from NYSE of the high frequency transactions for a stock in one day.

Table 4 lists some of the statistics of theses datasets. For an event $e = (y, t)$, the arrival times are available in each dataset. For the BookOrder dataset, mark $y$ is available in the data. However for the Twitter datasets, we extract $y$ as follows: $y = +1$ if $s \in (0, 1]$, $y = 0$ if $s = 0$ and $y = -1$ if $s \in [-1, 0)$, where $s$ is the sentiment of the message.

### F.2   Additional implementation details for base MTPP models

(i) RMTPP [17] [2]: Here we select batch size 64, initial learning rate 0.1 and $L_2$ penalty as $1e - 3$. (ii) THP [89] [3]: Here we select batch size 16, initial learning rate $1e - 4$ and $L_2$ penalty as 0 as the implementation contains no $L_2$ penalty. In addition, we used dropout rate 0.3, model depth 64, hidden layer size 128, head number 4 and layer number 4. (iii) SAHP [87] [4]. Here we select batch size 16, initial learning rate $1e - 4$ and $L_2$ penalty as $3e - 4$. In addition, we used dropout rate 0.1, model depth 16, hidden layer size 16, head number 8 and layer number 4. (iv) Linear model: For experiments with synthetic data, we used linear models[5][73] to build event simulator and parameter estimator.

### F.3   Compute resource

We used a single AMD Ryzen 7 3700X CPU with 32GB RAM machine, no graphical card is needed given all the models and data size in the experiments are moderate.

---

[2]https://github.com/musically-ut/tf_rmtpp

[3]https://github.com/SimiaoZuo/Transformer-Hawkes-Process

[4]https://github.com/QiangAIResearcher/sahp_repo

[5]https://github.com/sandeepsoni/MHP

# G Additional experiments with real data

| | Time prediction error in terms of MAE $\mathbb{E}(|t - \widehat{t}|)$ | | Mark prediction error $\Pr(y \neq \widehat{y})$ | |
|---|---|---|---|---|
| | TPP-SELECT | Base MTPP | TPP-SELECT | Base MTPP |
| Club | **3.78±0.00** | 3.80±0.00 | **0.49±0.01** | 0.79±0.02 |
| Election | **13.66±0.00** | 13.71±0.01 | **0.62±0.01** | 0.79±0.03 |
| Series | **1.98±0.01** | 2.13±0.02 | **0.62±0.00** | 0.67±0.01 |
| Verdict | **3.50±0.00** | 3.58±0.05 | **0.65±0.00** | 0.69±0.02 |
| BookOrder | **0.29±0.01** | 0.55±0.05 | **0.38±0.01** | 0.49±0.01 |

Table 5: Performance in terms of $\mathbb{E}(|t - \widehat{t}|)$, *i.e.*, the mean absolute error (MAE) for time prediction (left half) and the misclassification error $\Pr(y \neq \widehat{y})$ for the mark prediction (right half) for TPP-SELECT and the corresponding base MTPP model which is Self Attentive Hawkes Process (SAHP) in this case. Here, this base MTPP model is learned over the entire sequence of observed events. In all experiments, we considered 80% training and 20% test set.