# OpenReview forum: "Learning to Select Exogenous Events for Marked Temporal Point Process"
_NeurIPS.cc/2021/Conference — NeurIPS 2021 Poster_

### Official Review · Reviewer_Ls8f · 2021-07-14

**Rating:** 6
**Confidence:** 3

**Summary:**

This paper aims at identifying the set of exogenous events from a set of unlabeled events for learning marked temporal point process. This paper shows solving the exogenous event set selection problem, in conjunction with parameter estimation problem, can be achieved by maximizing a monotone and alpha-submodular set function and then proposes TPP-SELECT framework. The experiments on both synthetic and real data show the proposed method performs better than baselines.

**Limitations And Societal Impact:**

1. The current method doesn’t consider the triggering patterns among endogenous events, while there always exists a branch process among them. I wonder to know if this work can be extended to deal with such structure.
2. In Table 1, most of the mark prediction errors are larger than 0.5. Does there exist sample imbalance in those data set? Maybe ROC or AOC is a more suitable metric? I also suggest the authors test their method on data with different ratios of exogenous events to endogenous events.
3. There are some errors in Table 2: (1) Base MTPP achieves better results on Election data for time prediction error and on Club data for mark prediction error; (2) Base MTPP gets the same results with TPP-SELECT on BookOrder data for time prediction error. The authors should be more careful for the experimental results.
4. From Table 2, it seems base MTPP achieves comparable results with TPP-SELECT. However, I think base MTPP is much more efficient than TPP-SELECT. I suggest the authors provide further comparisons on convergence rate and run-time.

**Main Review:**

1. Overall, this paper is well organized and explains the framework clearly.
2. This paper targets at a very important and interesting problem, i.e., identifying the set of exogenous events without labels for marked temporal point processes. And it’s novel that applying submodular set function method to this problem, although it seems to be a relatively straightforward application.
3. The theoretical analysis is quite thorough.

Update:
Thank the authors for their comments. Most of my concerns are addressed. I will keep my score.

**Time Spent Reviewing:**

4 hours

---

> ### Author Response · Authors · 2021-08-10
> **Clarification about extending endogenous intensity in different setups and experiments**
>
> We would like to thank the reviewer for the suggestions. We will incorporate them in the revised version of the paper.
>
> > *”The current method doesn’t consider the triggering patterns among endogenous events, while there always exists a branch process among them. I wonder to know if this work can be extended to deal with such a structure.”*
>
> We believe that our method can be extended to those cases, where there is a branching structure between the endogenous events. More specifically, we can consider a more complex multivariate temporal point process following the work of [1] and can apply Eqs. (2--3) in [1] to compute the intensity function and the likelihood of the endogenous events. Then we would seek to extend our formulation on maximizing the underlying likelihood function w.r.t both the model parameters and the exogenous event set, which is likely to give similar setup under mild assumption on the intensity kernel $\phi$ in Eq. (2) in [1].
>
> [1] Shelton et al. 2018: Shelton, Christian R., Zhen Qin, and Chandini Shetty. "Hawkes process inference with missing data." In Thirty-Second AAAI Conference on Artificial Intelligence. 2018.
>
> > *”In Table 1, most of the mark prediction errors are larger than 0.5. Does there exist sample imbalance in those data set? Maybe ROC or AOC is a more suitable metric?”*
>
> Yes the data is imbalanced. Following the reviewer’s suggestions, we provide the AUC of our method and different baselines in the following table.
>
>
>   |    		|   TPP-Select	 |  EM 	| K-mean |  PCA	 | Fac-loc |
>  |----------------|---------------|--------------|----------------|-----------------|--------------|
> Club				|0.75|	0.77|	0.64|	0.56|	0.67
> Election			|0.70|	0.66|	0.59|	0.55|	0.60
> Series				|0.53|	0.52|	0.54|	0.52|	0.51
> Verdict				|0.64|	0.60|	0.53|	0.61|	0.58
> BookOrder			|0.71|	0.63|	0.63|	0.62|	0.64
>
> We observe that, for Election, Verdict and BookOrder datasets, our method outperforms the baselines.
>
> In the current version of the paper,  we avoided AUC because (i) classification accuracy is mostly adopted by the existing TPP works; and, (ii) except for BookOrder, the number of marks is 3 and therefore, the mark prediction involves *multiclass* classification. Hence, even for imbalance data, AUC may sometimes be undiscerning especially when size of samples with one mark is almost 50% whereas the other marks share the rest 50% of the points.
>
> >*"I also suggest the authors test their method on data with different ratios of exogenous events to endogenous events."*
>
> In the paper, we already analyzed this. Please refer to Figure 2 and subsequent discussion in lines 356--359. We observe that
> (i) our method shows poor performance when the fraction of exogenous events is either too small or too large and, (ii) a sweet spot of such a ratio exists, where our method provides best performance.
>
> > *”There are some errors in Table 2”*.
>
> We will rectify them in the revised version of the paper.
>
> > *”From Table 2, it seems base MTPP achieves comparable results with TPP-SELECT. However, I think base MTPP is much more efficient than TPP-SELECT. I suggest the authors provide further comparisons on convergence rate and run-time.*
>
> We provide the run time comparisons for Election dataset in the following table. We would like to highlight that, when $\mathcal{S}$ is reasonably high, we observe that RMTPP+TPP-Select is more efficient than RMTPP alone.
>
>
> |Run time for RMTPP alone (in s)	  | $\text{Size-}\mathcal{S}/\text{Size-}\mathcal{H} _T$	| Run time for  TPP-Select (RMTPP as base model) (in s)|
> |-------------------------------|-----------------------|------------------------------|
> 	16.31 |	0.1			| 16.95
>  "	|	0.3			| 16.06
>  "			  |	0.5			| 14.35
> "			  |	0.7			| 15.65
>
> We note that  the run time has two key components: (i) model training which predominantly involves estimation of parameters of endogenous model and (ii) iterative procedure in greedy algorithm. Now, as the size of exogenous events $|\mathcal{S}|$ increases, $|\mathcal{H} _T\backslash \mathcal{S}|$ decreases and therefore the time taken due to (i) decreases. However, as $|\mathcal{S}|$ increases,  the time taken due to (ii) increases due to larger number of iterations. Hence, we see the run time of our method  initially decreases with increase of $|\mathcal{S}|$ and then increases.

---

### Official Review · Reviewer_w3dd · 2021-07-15

**Rating:** 6
**Confidence:** 3

**Summary:**

This paper presents a method for identifying hidden labels indicating whether each event has been triggered by exogenous or endogenous influence. The parameter estimation problem of temporal point processes with event selection procedures is reformulated as a set function optimization problem. To solve this problem, the authors propose an approximate algorithm. The effectiveness of the proposed model is demonstrated using synthetic and real-world datasets.

**Limitations And Societal Impact:**

Please see comments (1) and (2) in my main review.

**Main Review:**

The major contribution of this paper is to bridge between the point process estimation problem and the submodularity and to develop a feasible optimization algorithm. This is technically solid. Thus, my opinion tends to accept. But there are some questions/suggestions as follows.

(1) In this paper, the exogenous events are regarded as those that are triggered by background intensities. They are defined as constants generated from probability distributions, as in Eq. (4). In a practical situation, one would like to model them as time-dependent functions such as periodic functions. Does the proposal allow such situations?

(2) Another option to efficiently infer triggering labels is the use of Gibbs sampler for latent variables (i.e., labels). Such methods are used for analyzing social and advertising effects, as in [R1, R2, R3]. They can also discern the events that are triggered by background intensity. These studies might be related to those that are cited in "Modeling external effects" of Section 1.3. It would be great to add a discussion about the differences and advantages of the proposal against these prior works.

[R1] T. Iwata, A. Shah, and Z. Ghahramani. Discovering Latent Influence in Online Social Activities via Shared Cascade Poisson Processes, KDD, 2013.

[R2] Scott W. Linderman, et al., Discovering Latent Network Structure in Point Process Data, ICML, 2014.

[R3] Y. Tanaka, et al., Inferring Latent Triggers of Purchases with Consideration of Social Effects and Media Advertisements, WSDM, 2016.

**Time Spent Reviewing:**

6 hours

---

> ### Author Response · Authors · 2021-08-10
> **Clarification about general form of exogenous model parameters and discussions about additional references**
>
> We would like to thank the reviewer for the suggestions. We will incorporate them in the revised version of the paper.
>
> > *"In this paper, the exogenous events are regarded as those that are triggered by background intensities. They are defined as constants generated from probability distributions, as in Eq. (4). In a practical situation, one would like to model them as time-dependent functions such as periodic functions. Does the proposal allow such situations?"*
>
> Yes, a time dependent function in such a situation is also allowed in our proposal. Please, note that the conditions and the results of our key technical theorems do not impose any strict restriction on the parameters of exogenous events. Therefore the key thesis of our proposal remains unchanged if we assume that $\mu$ and $q$ are time-dependent functions such as periodic functions.
>
> > *"Another option to efficiently infer triggering labels is the use of Gibbs sampler for latent variables (i.e., labels). Such methods are used for analyzing social and advertising effects, as in [R1, R2, R3]. They can also discern the events that are triggered by background intensity. These studies might be related to those that are cited in "Modeling external effects" of Section 1.3. It would be great to add a discussion about the differences and advantages of the proposal against these prior works."*
>
>
> We agree that these papers are related and we will cite them and add discussions about them in the revised version of the paper.  In particular, we would plan to write as follows:
>
>
> We note that [R1] aims to learn the effect of one user on the post of other users. Such a model can be used to quantify the external effect.  The proposal in [R2] aims to learn a probabilistic model which captures latent influence present in data using mutually exciting processes combined with a random graph model. Furthermore it infers the latent network structure from the noisy observations.  [R3] models dynamics of event sequence in the particular application of purchase data modeling. Specifically, it combines three factors, namely user preferences, influence from other users and media advertisements. These proposals learn their model using EM algorithm combined with Gibbs sampling method.  However, they do not provide an algorithm to explicitly  demarcate between endogenous and exogenous events.
>
>
> In this context, we acknowledge that one can build a new method inspired from the above methods, where we can assign a binary label $z$ to each event where $z=1$ indicates endogenous and $z=0$  indicates exogenous events and then, as the reviewer as pointed out, we can use Gibbs sampler mechanism for such latent variables.

---

### Official Review · Reviewer_mzmy · 2021-07-17

**Rating:** 7
**Confidence:** 4

**Summary:**

This paper studies a particular kind of temporal point process, with a focus on trying to identify exogenous events. The problem is formulated as a set function maximization, some theoretical results about monotonicity and alpha-sub-modularity are presented, a stochastic greedy algorithm (based on [38]) is proposed, and some experiments are conducted.

**Ethical Concerns:**

I have no ethical concerns.

**Limitations And Societal Impact:**

The authors should say much more about scope of their work; please see comments in my main review.

**Main Review:**

This paper makes some good theoretical contributions to the literature; it could have been (and could yet become) a stronger paper than its current form. I am left with the sense that there are innovative ideas here, but the results do not quite match up. I am open to upgrading my score after the rebuttal period, particularly if the authors can either convince me about some of their experiment-related choices, and/or make revisions. Let me begin with my major concerns about the paper.

Experiments:

I was enjoying the paper and was looking forward to giving it a strong accept, until I reached the experiments section, which made me change my mind instantly. I’m baffled by the authors’ choices. The paper is essentially about identifying exogeneous events, so I don’t understand why Section 5 delves into predicting next times and next labels. If they want to target these tasks, they are not considering the appropriate baseline temporal point process models.

My conjecture is that the authors felt they needed to use real data, and since real data does not usually come with labels about exogeneous / endogenous, they decided to tackle the prediction task. In the process, they have left out relevant baselines for this task. I believe they were better off focusing on synthetic data and showing that their method is able to identify exogeneous events better than relevant baseline approaches. I am not convinced their model is better than the state-of-the-art for prediction (it probably isn’t, which is fine).

I have some clarification questions about the experiments:

1. What exactly are the baselines in 5.1 doing? If I understand the task correctly, it is to predict the next time/label given the history of events. So, at any event occurrence in the test set, each model predicts the next time and event label. Is that correct? Are all of the baseline models generative models, i.e. can they generate new data? And why do they need to identify the exogenous events if the task is merely prediction? I would really like to understand what the authors were thinking here. It seems this entire experiment should be done with synthetic data, where the task is to identify exogeneous vs. endogenous events.
2. Why did the authors not compare with more recent models such as neural Hawkes, etc. for Table 2?

Literature:

For a paper with so many references, I’m surprised the authors have left out a lot of relevant literature. First, they have left out literature on outliers in temporal point processes, which is very closely related to the submission. I would be interested to hear how the authors think their work is different, and why they did not consider some of these as suitable baselines. I list a few papers below about outliers and also missing data (which I think should also be mentioned):

Event Outlier Detection in Continuous Time:
http://proceedings.mlr.press/v139/liu21g/liu21g.pdf

Detecting Anomalous Event Sequences with Temporal Point Processes:
https://arxiv.org/abs/2106.04465

Imputing missing events in continuous-time event streams:
http://proceedings.mlr.press/v97/mei19a/mei19a.pdf

Learning Temporal Point Processes with Intermittent Observations:
http://proceedings.mlr.press/v130/gupta21a.html

The authors mention multivariate point processes right at the end, but do not refer to the vast amount of literature on graphical models for temporal point processes. Some of this work is a generalization of the model considered in the paper. Here are some references:

Universal Models of Multivariate Temporal Point Processes:
http://proceedings.mlr.press/v51/gunawardana16.html

Proximal Graphical Event Models:
https://papers.nips.cc/paper/2018/file/f1ababf130ee6a25f12da7478af8f1ac-Paper.pdf

Also, I recommend citing the following paper, since it models exogenous events that affect state variables in a continuous-time Bayes net. Event-Driven Continuous Time Bayesian Networks:
https://ojs.aaai.org//index.php/AAAI/article/view/5725


Other comments/suggestions:

The term “externalities” is mentioned in the Abstract, with no explanation.

The description about the state-of-the-art in line 36 is inaccurate. Multivariate models do not consider these with “equal likelihood”. Please fix this line.

I recommend title case for section headings.

It may be worthwhile explaining the distribution choices in equation (4).

I found page 5 dense and hard to read. The work itself seems like a good contribution but its hard to follow. I suggest breaking it up further and adding more examples that clarify implications of some of the results. More discussion is needed here. Importantly, I think it’s very important to add more caveats about the specific MTPPs that are in scope for some of the results.

Equation (11) looks wrong for both parts. Lower and upper bounds are identical on the left, and t is used instead of y on the right.

Is k assumed known in algorithm 1? How is it determined in practice? Later on, it seems it has a major effect on results.

Can the authors say more about the subset sizes in step 3 of Algorithm 1? I understand that any subset is equally likely to be chosen. It seems that if small subsets are chosen, the algorithm is indeed efficient but performance could be poor. Did they try experiments where they controlled the subset size?

Figure 1 is not as easy to read as the authors perhaps intend. I could not spot the great results that the authors claim. I see both greener and redder as my eyes scan to the right. Perhaps a table with classification numbers would help here?

There is a mistake in lines 315-317: 7 datasets are implied here but only 5 were used.

There are several typos/grammatical errors. Here are some: line 120 (allows), 171 (variety), 184-186 (entire sentence needs editing), 332 (hyper-parameters), 335 (on in), etc.

**Time Spent Reviewing:**

5

---

> ### Author Response · Authors · 2021-08-10
> **Explanation on baselines and other minor comments**
>
> We would like to thank the reviewer for the suggestions. We will incorporate them in the revised version of the paper.
>
> > *”What exactly are the baselines in 5.1 doing? It is to predict the next time/label”*
>
> We believe that there is some misunderstanding about the baselines in Section 5.1.  These baselines (EM, K-means, PCA, Fac-loc) as-it-is do not predict the next time or mark. Similar to our proposed method, they are simply unsupervised event classification mechanisms which are used to classify endogenous and  exogenous events from the set of unlabeled training events.  However, to assess the quality of these baselines,  we use these endogenous and exogenous events provided by  different baselines, to train the *same* endogenous TPP model (RMTPP in our case, as noted in the caption of Table 1 and line 342) and the *same* exogenous TPP model (Eq. 4). Finally, these different trained versions of the same TPP model ---  which are learned using different partitions of endogenous and exogenous events given by different baselines in Section 5.1---  are used to predict the next event. We believe that the difference in the resulting predictive performances can be associated with the difference in quality of the baselines, since the underlying TPP model  remains the same.
>
> In a nutshell,  we highlight that unlike any TPP model,  the baselines as-it-is are not time or mark predictors, they are simply event selectors/classifiers. However, in absence of ground truth labels, we evaluate the quality of these baselines by measuring the predictive power of one standard TPP model trained on the different choices of endogenous and exogenous events, obtained from different baselines. Hence, the quality of demarcating endogenous and exogenous events by any event selection method is reflected via the predictive performance of the underlying TPP model.
>
> More formally, we proceed as follows. Given a set of training events $\mathcal{H}$, we use different baselines $\mathcal{B} \in $ (EM, K-means, PCA, Fac-loc) to split  $\mathcal{H}$ into different partitions of endogenous $\mathcal{H} \backslash \mathcal{S} _{\mathcal{B}}$  and exogenous events $\mathcal{S} _{\mathcal{B}}$. Note that, different baselines $\mathcal{B}$ give us different partitions  $(\mathcal{S}  _{\mathcal{B}} , \mathcal{H} \backslash \mathcal{S}  _{\mathcal{B}} )$.  Now, we consider one same pair of endogenous TPP models $[(\lambda  _{\theta}, m _{\theta})$ and $(\mu,q)$ for endogenous and exogenous TPP] and train it with these exogenous-endogenous event partitions $(\mathcal{S}  _{\mathcal{B}} , \mathcal{H} \backslash \mathcal{S}  _{\mathcal{B}} )$  given by the baseline $\mathcal{B}$.  In this process, we obtain different parameters for TPP models, one for each partition that was used to train it.  Each set of parameters gives us a new TPP model of exactly the same kind, but with different predictive performance.
>
> > *”Why not baselines are from TPP models”*
>
> Our proposal does not provide a new TPP model--- it provides a method for exogenous event selection while learning any given TPP model. Hence, evaluation of our event selection method requires comparison with different event selection methods while keeping the underlying TPP model the same. For example, if we train RMTPP on different choices of endogenous and exogenous events given by different baselines, the different predictive performances of RMTPP can be connected with classification quality of different baselines.  On the other hand,  if we use different TPP models, it is very difficult to conclude whether the difference in predictive performances is due to difference in model or event selector.  Hence, we need to keep the underlying TPP model the same across different baselines to properly evaluate the impact of the baselines on model training and the predictive performance.
>
>
> That being said, we agree that comparing different TPP models with the same event selector would also reveal how useful our method is across a wide variety of TPP models. We will add a discussion about it  in the revised version of the paper.
>
>
> > *”Why we identify exogenous events if the task is merely prediction?”*
>
> To obtain accurate estimates of the parameters of the underlying endogenous and exogenous components (Eqs. 3 and 4 in the paper) of the TPP model, these components should be trained with correct partitions of endogenous and exogenous events respectively. Therefore, if the exogenous event selection process is poor, it can label a large number of endogenous events as exogenous events. This would lead to training of the exogenous model with a large number of endogenous events and vice-a-versa.  It would further result in poor performance both at the model training and  event prediction. Hence, even for prediction tasks, an accurate selection of exogenous events is a critically important step.
>
>
> > *”entire experiment should be done with synthetic data, where the task is to identify exogenous vs. endogenous events”*
>
>
> As noted by the reviewer, real data does not have endogenous and exogenous labels associated with events and therefore, a direct evaluation is extremely difficult.  On the other hand, one of the key motivations of an event selection strategy is to improve  the performance of the model during training and inference.  Hence, we believe that the quality of the different event selection methods can be reasonably compared by evaluating the predictive ability of the same TPP model trained with the different endogenous and exogenous event sets given by these methods.
>
> That being said, we would like to acknowledge that experiments with synthetic data would improve the clarity. Therefore, as suggested by Reviewer 8RFc, we performed experiments where we assumed that all available events in real data are endogenous and then we sampled some fake events generated using our exogenous model. Finally, we applied our method and the baselines in Section 5.1 to classify these generated exogenous events.  We summarize the classification accuracy of the exogenous events here.
>
>
>  |		| TPP-Select	 |      EM	| 	K-means	 |	 PCA	 |   Fac-loc|
>  |-----------------------|-------------|-------|-------|-------|-------|
> Club	|	0.42	 |0.41	|0.31		|0.09	| 	0.28
> Election	|0.48		|	0.29 	|	0.46 		|	0.15	| 0.28
> Series	|	0.17 		|	0.43	|	0.26 		|	0.09     | 0.28
> Verdict		| 0.32		|	0.25	|	0.29		|	0.05	| 0.28
> BookOrder	| 0.61		|	0.44	|	0.27		|	0.28	| 0.28
>
>
> We observe that TPP-Select outperforms the baselines in most cases. However, we observe that in the Series dataset, EM outperforms all other methods.
>
> > *”more recent models such as neural Hawkes”*
>
> Apart from RMTPP, we also performed experiments with two other  TPP-models, (i) Transformer Hawkes Process (Zuo et al 2020, THP, please refer to Table 2 in the main paper) and (ii) Self Attentive Hawkes Process (Zhang et al 2020, SAHP, please refer to Table 4 in supplementary material). Note that both THP and SAHP are more recent than the Neural Hawkes Process (Mei et al 2017, NHP).
>
> The original implementation of Mei et al. 2017 paper was written in Theano. On the other hand, the pytorch version of NHP was integrated in another work which is on imputing missing events.  While this repository contains the NHP model, we are unable to find any routine that is customizable for the prediction task using the trained model in the context of our setup.  Hence,  we found it   difficult to adapt their code to our context. Due to these reasons, we chose THP and SAHP which are more recent as well as easier to adapt to our setup. However, if the reviewer wishes, we can also perform experiments with the Neural Hawkes process and include it in the revised version of the paper.
>
> > *”distribution choices in Eq. (4)”*
>
> Here, our objective was to generate random variables for exogenous intensity ($\mu$) and mark distribution ($q$) which should satisfy that $\mu\in [0,\infty) $ and $q\in [0,1]$. We thought that gamma and beta distribution may be the common distributions which could model these quantities.
>
> > *”Equation (11) looks wrong for both parts”*
>
> Lower and upper bounds are not identical on the left. On the left eq. of (11), the exponent in LHS has a negative sign. Yes, y should be on the right instead of t.
>
> > *”How is k determined in practice?”*
>
> As we mentioned in line 332, we treat k as a hyperparameter  and obtain it using cross validation on the validation set.
>
> > *”Can the authors say more about the subset sizes in step 3 of Algorithm 1?”*
>
> In the following table, we summarize the effect of the subset size on both the classification accuracy and the time taken to run Algorithm 1 on our synthetic dataset.
>
> | $\text{Size-}V/\text{Size-}H$  | Accuracy	| Time (s)|
> |---------------|------------|------------|
> 0.1|0.806 | 81.73
> 0.3	| 0.813  | 233.68
> 0.5  | 0.819   | 393.30
> 0.7| 0.806  | 552.81
>
>
> We used k = 0.3 |V|, where k is the maximum size of the exogenous set. Note that even with a smaller size of V, we observe a significant accuracy. This is because we choose different V at each step of Algorithm 1 and as a consequence, the overall exploration space of searching for candidate set S is actually large, while the algorithm is still efficient.
>
> > *”more caveats about the specific MTPPs that are in scope for some of the results”*
>
> We discussed some  MTPPs in the context of applicability of the current results in Appendix C.3. We will elaborate this discussion further.
>
> > *”Figure 1 is not as easy to read [...] Perhaps a table with [...] numbers”*
>
> We provide the table here. We will also include it in the revised version of the paper.
>
> | Iterations	 | Acc. (exo events)	| Acc. (endo events) | Total Acc.|
> |-------|--------|--------|-----------|
>  k/3		 |0.866| 0.823|0.826
> 2k/3		 | 0.700	| 0.855|	0.831
>  k		 |0.577|0.874|0.806
>
>
> We will address all other minor comments, rectify the typographical errors and discuss the suggested papers in the final version of the paper.

---

> > ### Comment · Reviewer_mzmy · 2021-08-22
> > **Follow-up confirmation question about experiments**
> >
> > Thanks to the authors for their detailed response with clarifications, which I appreciate. Could they please confirm my understanding of the experiment in Table 1, which I summarize below:
> >
> > - First, one determines which of the events are endo/exo in the train set. For this, TPP-Select (Algorithm 1) is run along with others like EM, K-means, etc.
> > - Then, one takes a neural model (like RMTPP) and learns a model for the endo events using train/valid sets; TPP-Select does this already but this needs to be done separately for the other methods. The exo events are treated as homogeneous Poisson, for which the parameter is learned separately. This provides a (combined) generative model with independent processes for the endo/exo events for each method, i.e. column of Table 1. Each method will result in a different generative model because the chosen endo/exo events could be different in the train set.
> > - Each (combined) generative model is asked to perform a prediction task on the test set. Specifically, it must predict the next event (time as well as label) in the test set, given the history. This is done for every observed event in the test set, including the first event.
> >
> > Is the above correct?

---

> > > ### Author Response · Authors · 2021-08-23
> > > **Confirmation about the clarifications of experiments**
> > >
> > > We thank the reviewer for getting back and confirm that the points mentioned by the reviewer are correct.

---

> > ### Comment · Reviewer_mzmy · 2021-08-28
> > **Comparison with literature**
> >
> > A couple more questions for the authors. I'm curious to learn about:
> > 1) Why they did not include some of the "outliers in point processes" literature, some of which I mentioned in my review (saying that they were not aware of the literature is of course a reasonable answer).
> > 2) How they think their work differs from this literature.

---

> > > ### Author Response · Authors · 2021-08-28
> > > **Clarification re. literature**
> > >
> > > > *Why they did not include some of the "outliers in point processes" literature, some of which I mentioned in my review (saying that they were not aware of the literature is of course a reasonable answer)*.
> > >
> > > The two outlier detection papers did not come to our attention until recently. Although the paper titled “Event Outlier Detection in Continuous Time” was available on arxiv, it got published in ICML 2021.  Moreover, we would like to highlight that the other paper titled “Detecting Anomalous Event Sequences with Temporal Point Processes” *came in arxiv  a few days  after NeurIPS supplementary deadline*. We will add a discussion in the revised version of the paper to highlight the differences from our work.
> > >
> > > > *How they think their work differs from this literature*
> > >
> > > (1) *Differences with “Detecting Anomalous Event Sequences with Temporal Point Processes”*
> > >
> > > This paper considers detecting anomalous MTPP *sequences* $\{X_a\}$ from a set of sequences $D = (X _1, X_2, …, X_n)$. Hence, their goal is to label the *entire sequence* as anomalous or non-anomalous. In contrast, our work aims to find the *exogenous events* $S$ from *one given sequence $X$*. Hence, the goals of our paper and their paper are different.
> > >
> > > As a consequence, the adopted methods to solve these problems are also different. In particular, they follow an hypothesis testing methodology to detect anomalous sequences. On the other hand, we give an approximation algorithm for the event selection mechanism which selects exogenous events from a given sequence of events.
> > >
> > > (2) *Differences with “Event Outlier Detection in Continuous Time”*
> > >
> > > This paper considers a semi-supervised setting.  In their setup, they assume that they have access to data without outliers, so that the underlying MTPP model can be trained on this clean data  (Check the last paragraph “Semi-Supervised Detection” in Section 2 (i.e. Related work) and the discussion below Eq. (1) in their paper ).
> > >
> > >
> > > In contrast, our work applies to an unsupervised setting, where the training data also contains the exogenous events (in addition to usual endogenous events), which we believe to be more realistic and challenging. Thus, our setup is different and hence, we adopt completely different method to solve our task.

---

> > > > ### Author Response · Authors · 2021-08-31
> > > > **Re. clarifications to reviewer's response**
> > > >
> > > > We would like to thank the reviewer for the discussions. It would be great if the reviewer let us know if our responses clarified all the questions/concerns of the reviewer and we will be happy to provide any further clarifications.

---

> > ### Comment · Reviewer_mzmy · 2021-09-02
> > **Updated comments**
> >
> > Thanks again to the authors for their responses. I don't think the prediction experiments reveal much that is new to the field - they indicate that it can be better to use simpler models for some event types; this is well known for multivariate point process models. However, my initial reaction was that the prediction experiments were completely meaningless - I realize this is not the case and it's good that the authors clarified this aspect. I also like some additional experiments that the authors conducted, which in my view are more aligned with their core idea around recognizing endo/exo event types. For these reasons, I have increased my score.
> >
> > I hope the authors will add relevant literature, provide necessary clarifications and revise the paper accordingly. I strongly suggest that the authors minimize the prediction related experiments in their paper, particularly since they have other results that tie in closer to the endo/exo idea, but of course that is their prerogative.

---

### Official Review · Reviewer_8RFc · 2021-07-17

**Rating:** 8
**Confidence:** 4

**Summary:**

The authors propose an approach to separate exogenous and endogenous events generated by marked temporal point processes (MTPPs). In this context, an event consists of a timestamp and a mark, which is assumed to be a category or class. Many MTPPs are self-exciting or self-inhibiting so that new events are generated due to the influence of previous events--these are endogenous events. Other events are considered to be exogenous events. Given a set of events, the authors proposed approach attempts to select a subset of exogenous events using a stochastic greedy approach. Along the way, they establish some important theoretical properties regarding monotonicity and submodularity of MTPPs that allows them to derive an approximation guarantee for their algorithm. They also demonstrate reasonable empirical results on simulated and real data sets.

**Limitations And Societal Impact:**

Both are discussed in the supplement. The discussion of limitations is somewhat brief and should be expanded upon. Another limitation that I noticed is that the proposed algorithm is a hard decision algorithm--unlike the EM algorithm baseline used in this paper or the variational inference approach in Li and Zha (2013), which output a probability that a given event is exogenous.

**Main Review:**

**Originality: High**
While the problem of separating exogenous and endogenous events in temporal point processes (TPPs) or MTPPs is not new, the proposed approach completely changes the way we think about exogenous and endogenous events in MTPPs. The typical approach, as the authors mention, is to assume that the conditional intensity function $\lambda(t)$ is the sum of exogenous and endogenous components, which are then treated as a single MTPP. This sum is then treated as the starting point to separate exogenous and endogenous.

The authors instead propose to think about exogenous and endogenous as separate MTPPs whose events are interleaved so that they look like they are generated from a single MTPP. They demonstrate the relationship between their approach and the usual sum approach in Appendix B.1. They exploit their new modeling approach to better classify exogenous and endogenous events.

The discussion of related work is fairly comprehensive. One missing reference is Li and Zha (2013), who use a mixture of linear Hawkes processes to classify events as being from different classes.

Reference:
Li, L., & Zha, H. (2013). Dyadic event attribution in social networks with mixtures of hawkes processes. Proceedings of the 22nd ACM international conference on Conference on information & knowledge management - CIKM '13, 1667-1672. doi:10.1145/2505515.2505609

**Quality: Moderate**
The quality of the theoretical results is higher than the experimental results, in my opinion (although I did not carefully check proofs). Theorems 3-6 enable the authors to derive the approximation guarantee for their algorithm (Theorem 7), and as the authors note, it may be of independent interest.

The experimental results presented suggest that the proposed TPP-Select algorithm works, but does not establish superiority of TPP-Select compared to baselines at the proposed task: separating exogenous and endogenous events. The authors opt to present only qualitative simulation results while deferring quantitative results to Appendix E. I think this is a mistake. Furthermore, the authors don't compare error rates of different approaches with respect to classification of exogenous/endogenous; instead, they only evaluate prediction error rates.

I would suggest 2 additional experiments:
- On your current synthetic data, measure also classification error for TPP-Select and compare it to the EM and k-means baselines, which also attempt to identify the exogenous event set.
- On your real data experiments, assume that all events are endogenous and then inject (interleave) some fake events generated according to your exogenous model. Then evaluate classification error with respect to these fake events.

**Clarity: Moderate**
The quality of the presentation is decent, with lots of minor issues as I list below. More importantly, I had to read through almost the entire paper before I understood the distinction between exogenous and endogenous that the authors are making. I would suggest showing a linear Hawkes process as an illustrative example in Section 2.2 to clearly show that $\mu$ corresponds to the exogenous component.

The choice of experiment results to include in the main paper and supplement could also be improved. I think Figure 1 and 2 should go to the supplement while something similar to Figure 3 should go into the main paper (see discussion above for improvements to the simulation experiments).

Minor issues:
- Definition of $\lambda(t)$ in (1) is a bit sloppy--it should be the limit of the stated quantity as $dt \rightarrow 0$.
- In equation (9), the subscript $\lambda$ on $\theta$ and $\kappa$ should be replaced with the black dot that is being used elsewhere.
- Line 211: MTTPs -> MTPPs
- Line 295: missing end ) in the first exp function

**Significance: High**
This paper presents a novel way of thinking about TPPs. Given the large amount of research being done on TPPs recently (particularly non-linear and neural TPPs), I think this has the potential to be very impactful.


**Time Spent Reviewing:**

2 hours

---

> ### Author Response · Authors · 2021-08-09
> **Results of additional experiments and response to other minor comments**
>
> We would like to thank the reviewer for the positive reviews, which we would incorporate in the revised version of the paper. We address the concerns and the suggestions of the reviewer in the following.
>
>
>
>
> > *"The authors opt to present only qualitative simulation results while deferring quantitative results to Appendix E. I think this is a mistake"*
>
> Since we already added quantitative results for experiments with real data, we initially decided to defer the quantitative results for the synthetic data in the appendix. However, as the reviewer has suggested, we will bring back quantitative results for the synthetic data in the main part of the revised version of the paper.
>
>
> > *"Furthermore, the authors don't compare error rates of different approaches with respect to classification of exogenous/endogenous; instead, they only evaluate prediction error rates"*
>
> Since the real data we experimented with does not come with labels, we could not evaluate our method in terms of classification accuracy, but resorted to evaluate in terms of prediction error of the underlying TPP model.
> However,  during the rebuttal period, we performed the two additional experiments suggested  by the reviewer, which indeed led to a more direct evaluation of our proposal. We summarize the results as follows:
>
>
> > *"On your current synthetic data, measure also classification error for TPP-Select and compare it to the EM and k-means baselines, which also attempt to identify the exogenous event set"*
>
> We have performed this experiment during the rebuttal period. They are summarized as follows.
>
>  |TPP-Select  |  EM    |    K-means | PCA   |  Fac-loc|
>  |--------------|-------------|-------|-------|-------|
>       	  |0.806         | 0.554 |      0.620     | 0.562 |  0.299 |
>
> We observe that TPP-Select outperforms the baselines by a significant margin.
> > *"On your real data experiments, assume that all events are endogenous and then inject (interleave) some fake events generated according to your exogenous model. Then evaluate classification error with respect to these fake events."*
>
> We have performed this experiment during the rebuttal period. They are summarized as follows.
>
>
>  |		| TPP-Select	 |      EM	| 	K-means	 |	 PCA	 |   Fac-loc|
>  |-----------------------|-------------|-------|-------|-------|-------|
> Club	|	0.42	 |0.41	|0.31		|0.09	| 	0.28
> Election	|0.48		|	0.29 	|	0.46 		|	0.15	| 0.28
> Series	|	0.17 		|	0.43	|	0.26 		|	0.09     | 0.28
> Verdict		| 0.32		|	0.25	|	0.29		|	0.05	| 0.28
> BookOrder	| 0.61		|	0.44	|	0.27		|	0.28	| 0.28
>
> We observe that TPP-Select outperforms the baselines in most cases.   However, we observe that in Series, EM outperforms all other methods.
>
> > *"I would suggest showing a linear Hawkes process as an illustrative example in Section 2.2 to clearly show that μ corresponds to the exogenous component"*
>
> We would add such an illustration in the revised version of the paper.
>
>
> > *"I think Figure 1 and 2 should go to the supplement while something similar to Figure 3 should go into the main paper"*
>
> We will present the results of the additional experiments suggested by the reviewer in the similar manner to Figure 3 and present them in the main part of the revised version of the paper.
>
> > *"Another limitation that I noticed is that the proposed algorithm is a hard decision algorithm--unlike the EM algorithm baseline used in this paper or the variational inference approach in Li and Zha (2013), which output a probability that a given event is exogenous"*
>
> Yes, we provide a hard decision algorithm, whereas, the algorithm proposed by Li and Zha (2013) is a soft decision algorithm. We will expand the limitation sections and elaborate with a discussion on this.
>
> We will address all the minor issues, add the suggested citations and elaborate the limitations and societal impacts in the light of the reviewer’s suggestions in the revised version of the paper.

---

> > ### Comment · Reviewer_8RFc · 2021-09-02
> > **Thanks for the new experiment results!**
> >
> > I thank the authors for providing the new experiment results. Indeed, the inclusion of these experiments would improve the weakest part of this paper currently--the experimental evaluation. I continue to support this paper.

---

### Decision · Program_Chairs · 2021-09-27

**Decision:**

Accept (Poster)

**Comment:**

The submission provides useful theory and experimental support for methods to label Marked Temporal Point Processes. There was extensive engagement between the authors and reviewers, and the reviewers came away well satisfied by the extended experiments. We ask that the authors please update the paper to incorporate as much of the useful additional exposition as possible, updating the main paper where possible or including supplemental material where appropriate.